# Genome anchoring, retention, and release by neck proteins of *Staphylococcus* phage 812
Zuzana Cieniková [1], Jiří Nováček [1], Marta Šiborová [1,3], Barbora Popelářová [1,2], Tibor Füzik [1], Tibor Botka [2], Martin Benešík[2], Pavol Bárdy [2,4], Roman Pantůček [2] & Pavel Plevka [1] ✉

The virion of *Staphylococcus* phage 812 is formed by a capsid and a contractile tail joined together by neck proteins. The neck proteins are crucial for virion assembly, DNA packaging, and the regulation of genome release, but their functions are not completely understood. Here, we show that the neck of phage 812 consists of portal, adaptor, stopper, tail terminator, and two types of decoration proteins. A dodecameric DNA-binding site at the surface of the portal complex anchors the phage genome inside the capsid. The adaptor complex induces a local B-to-A form transition of the DNA in the neck channel that could slow or pause genome translocation during ejection. The central channel of a stopper complex that is not attached to the tail terminator complex is closed by gating loops. In contrast, in the phage 812 virion, the gating loops are in an open conformation, and the DNA extends into the tail. The structure of neck proteins is not affected by tail sheath contraction. Therefore, the expulsion of tail tape measure proteins triggers the genome release.

Lytic phage 812, strain K1/420, and closely related kayviruses from the family *Herelleviridae* infect a wide range of *Staphylococcus aureus* strains. Kayviruses are promising candidates for phage therapy against human infections caused by *S. aureus*, a Gram-positive pathogen[1–3]. The virion of phage 812 is formed by an icosahedral capsid enclosing a 150 kbp-long, AT-rich dsDNA genome, and a double-layered contractile tail terminated by a baseplate[2,4]. The tail, possessing helical symmetry, is attached to the capsid by the neck. The neck of long-tailed phages is formed by a dodecamer of portal proteins, a dodecamer of adaptor proteins, a hexamer of stopper proteins, and a hexamer of tail terminator proteins[5–10]. Adaptor and stopper proteins are collectively called head completion or head-to-tail joining proteins. Tail terminator is also called tail completion or tail-to-head joining protein.

The neck is involved in numerous functions during the phage infection cycle. The portal complex nucleates capsid assembly[11,12]. In the completed prohead, the portal dodecamer occupies the position of a pentamer of capsid proteins at one of the icosahedral vertices. During genome filling, the packaging terminase complex docks onto the portal complex and threads genomic DNA through its central channel into the capsid[13–15]. In phage T4, the packaging starts with the insertion of about 500 bp of the genomic leading end into the capsid and its anchoring to the portal complex[16]. The

crown domain of the portal complex located inside the capsid was proposed to function as a sensor for the completion of genome packaging in phages SPP1 and P22[17,18]. After the termination of the packaging and detachment of the terminase, head completion proteins assemble at the portal complex and ensure genome retention in tailless proheads[19,20]. Multiple mechanisms of genome retention in virions of long-tailed phages have been described: in the *Bacillus* phage SPP1, the genome is held inside the neck channel by the stopper proteins[6], whereas in the *Staphylococcus* phage 80α it is stopped by the tail completion protein located inside the neck channel[21], and in the phages Pam3 and λ by the tail tape measure proteins at the neck-to-tail junction[9,22]. The DNA double-strand is inherently asymmetric and flexible, and thus can only be characterized using localized structural studies bypassing symmetry averaging, which are challenging even with state-of-the-art equipment and methodologies[21,22].

Here, we present cryo-electron microscopy (cryo-EM) structures of the neck of the phage 812 virion and genome release intermediate, as well as the crystal structure of a hexamer of stopper proteins. We show that a positively charged belt at the surface of the portal complex binds the DNA inside the capsid. In the absence of attachment to the tail terminator complex, the central channel of the stopper protein hexamer is closed by gating loops. In the phage 812 virion, where the stopper complex is attached to the tail

[1]CEITEC Masaryk University, Kamenice 753/5, 625 00, Brno, Czech Republic. [2]Department of Experimental Biology, Faculty of Science, Masaryk University, Kamenice 753/5, 625 00, Brno, Czech Republic. [3]Present address: Novo Nordisk Foundation Centre for Protein Research, Faculty of Health and Medical Sciences, University of Copenhagen, 2200 Copenhagen, Denmark. [4]Present address: York Structural Biology Laboratory, Department of Chemistry, University of York, York, YO10 5DD, UK. ✉e-mail: pavel.plevka@ceitec.muni.cz

terminator complex, the central channel is open and DNA protrudes into the tail. The portal and adaptor complexes bend the DNA inside the neck channel and induce its local transition to an A-form. These observations led us to propose a mechanism for how the neck proteins of *Herelleviridae* phages anchor, retain, and modulate the release of the phage genome.

## Results and Discussion

### Neck structure of phage 812 virion

The cryo-EM structures of the phage 812 neck solved with imposed sixfold and twelvefold symmetries both achieved a resolution of 4.2 Å (Fig. 1, S1–S5 and Table S1). The neck of phage 812 consists of four types of proteins forming the channel: a dodecamer of gp91 portal proteins, a dodecamer of gp96 adaptor proteins, a hexamer of gp97 stopper proteins, and a hexamer of gp99 tail terminator proteins (Fig. 1A and Tables S2–S4). The outer surface of the neck is decorated with gp164 stopper decoration proteins and gp56 terminator decoration proteins, which are organized as hexamers of homodimers (Fig. 1A). The density of putative neck whisker proteins surrounding the adaptor complex was not sufficiently resolved to enable structure determination and gene product identification (Fig. 1B).

### Portal complex

The portal complex of phage 812 has the shape of a spool, with an inner 30 Å-wide channel. The portal protein can be divided into wing, crown, stem, and clip domains (Fig. S6A and Table S3)[5]. The wing and crown domains are inside the capsid, the stem domains traverse the capsid, and the clip domains

are inserted into the adaptor complex outside the capsid (Fig. 1A–C). The crown domains form a conical funnel that opens towards the center of the head. In the phage 812 virion, three sections of the dsDNA genome encircle the crown and the wing domains (Fig. 1B). The outer surface of the wing domain has an overall positive charge (Fig. S7 and Table S5). This feature is conserved in phages from the *Herelleviridae* family (Fig. S8), and was also found in herpesviruses (Fig. S9)[23]. In contrast, the surfaces of the wing domains of phages from other families are mostly negatively charged (Fig. S9). Portal complexes were shown to interact with the packaged DNA and thus influence genome topology and retention[16,18]. The role of the positive charges of wing domains of phage 812 portal proteins in DNA binding is discussed below.

### Adaptor proteins

The dodecamer of adaptor proteins has the overall shape of a funnel (Fig. S6B). The adaptor protein of phage 812 is composed of an α-helical core, a β-hairpin, and a β-sandwich domain. The channel inside the adaptor complex broadens to 80 Å in diameter. The channel chamber is walled by the α-helical cores (Fig. 1B, C). β-hairpins of the twelve adaptor proteins assemble into a cylindrical tube with a diameter of <30 Å, inserted into the stopper complex (Fig. 1B, C). The diameter of the adaptor β-hairpin tube channel is remarkably conserved among both sipho- and myophages with known structures[8–10,21,22,24–26], and is probably optimized for the translocation of straight B-form dsDNA. The β-sandwich subdomains of phage 812 adaptor proteins are positioned on the outer surface of the neck and are in contact with the capsid (Fig. 1B, C). The adaptor protein of phage 812 is structurally

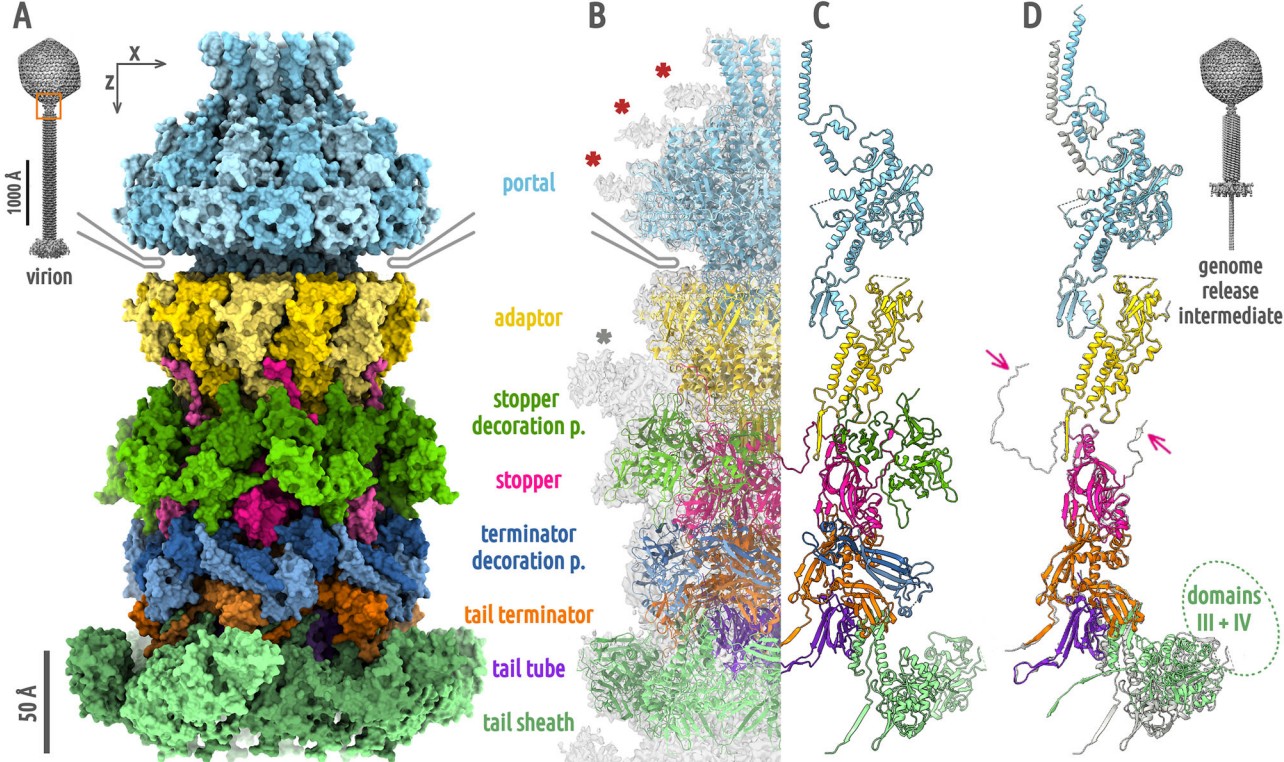

**Fig. 1 | Structure of phage 812 neck. A** Neck proteins of phage 812 virion are shown in surface representation: portal in light blue, adaptor in gold, stopper in magenta, tail terminator in orange, stopper decoration in forest green, terminator decoration in steel blue, tail tube in violet, and tail sheath in light green. The capsid is indicated schematically in gray. The neck region is outlined with an orange box in the overview of the virion. **B** Neck proteins of the virion are shown in cartoon representation, with the cryo-EM map reconstructed with imposed C6-symmetry (EMD-18462) shown as semi-transparent light gray surface. The densities of three DNA rings encircling the portal complex are marked with red asterisks, and the density of putative neck whisker proteins with a gray asterisk. **C** Cartoon representation of monomers of

neck proteins of the phage 812 virion. Domains III and IV of the tail sheath protein are shown as semi-transparent. **D** Comparison of the structure of neck proteins in virion (light gray) and in genome release intermediate (in color). Decoration proteins were not present in the genome release intermediate and are not shown for the virion. Magenta arrows indicate the N- and C-termini of the stopper protein resolved only in the virion. Domains III and IV of the tail sheath protein from the neck-proximal disc are not resolved to a high resolution in the structure of the phage 812 genome release intermediate. Their approximate position is indicated by a light green dotted oval. The inset shows the overall structure of the genome release intermediate.

similar to its homolog in the *Mycolicibacterium* phage Mycofy1, where an α-helical core, a β-hairpin, and a β-sandwich domain were also described[26].

## Stopper proteins

The hexamer of stopper proteins of phage 812 is shaped as a ring, with extensions stretching towards the adaptor complex (Fig. 1A). The stopper protein is composed of a β-barrel core, a tetra-cysteine loop, a β-hairpin gating loop, and an external β-sandwich domain (Fig. S6C). Six β-barrel cores delimit the neck channel to 40 Å in diameter. The β-hairpin gating loops in the phage 812 virion line the channel and are sandwiched between hairpin loops of tail terminator proteins. The twelvefold symmetry of the adaptor complex is preserved at the interface with the hexamer of stopper proteins, as the stopper protein forms similar intermolecular contacts with a pair of neighboring adaptor proteins (Fig. S10). The phage 812 stopper protein has a close structural resemblance to its homologs from *Pseudomonas* phages E217 and Pa193, which however lack the external β-sandwich domain[10,25].

## Tail terminator proteins

The tail terminator complex of phage 812 has the shape of a six-armed clamp with an inner 40 Å-wide channel (Fig. 1B, C). The tail terminator protein is formed by a core domain composed of a β-sheet flanked by α-helices and a β-sandwich domain (Fig. S6D). The core β-sheets line the central channel and are surrounded by α-helices on the outside (Fig. 1C). The β-sandwich domain binds to tail sheath proteins (Fig. 1A, B). The tail terminator protein of phage 812 bears a structural similarity to its homologs in phages E217 and Pa193, apart from the β-sandwich domain present solely in phage 812[10,25].

## Tail tube and tail sheath proteins

The structures of the phage 812 tail tube and tail sheath were determined previously to resolutions of 9 and 6 Å, respectively[4]. Here, we present the structures of neck-adjacent tail tube and tail sheath proteins determined to a resolution of 4.3 Å (Fig. S3C). The tail tube protein consists of a β-sheet domain, six of which build a β-barrel that forms the tail channel (Fig. 1A–C, S6H). A hexamer of tail sheath proteins surrounds each ring of tail tube proteins (Fig. 1A, B). The tail sheath protein can be divided into domains I–IV[27]. Domains I and II are adjacent to the tail tube. Domains III and IV are located at the outer face of the tail sheath (Fig. S6G). Domains III and IV are less resolved in the cryo-EM reconstruction due to their flexibility, and were interpreted using predicted structures.

## Stopper and terminator decoration proteins

The decoration proteins of phage 812 neck form elongated dimers spanning, in a zigzag pattern, the adaptor-stopper junction (stopper decoration protein) and stopper-terminator junction (terminator decoration protein) (Fig. 1A). The stopper decoration protein is composed of an N-terminal globular domain containing a β-sheet flanked by two α-helices, and a C-terminal β-barrel domain (Fig. S6E). The terminator decoration protein contains 15-residue-long N- and C-terminal antiparallel β-strands and a small globular central domain (Fig. S6F). In the phage 812 virion, the stopper decoration proteins interact with a density of putative neck whiskers and probably enable their attachment (Fig. 1B). We hypothesize that the decoration proteins reinforce the interfaces between adaptor and stopper, and stopper and terminator complexes, since phage 812 particles that had lost the neck decoration proteins often have deformed or broken necks (Fig. S11A). It is also possible that the neck whisker and decoration proteins play a role of chaperones that are important for a successful self-assembly of phage 812 virions.

## Stopper complex ensures genome retention in the capsid before tail attachment

After the completion of phage genome packaging and dissociation of the packaging terminase, adaptor and stopper proteins attach to the portal complex[19]. The resulting genome-filled assembly intermediate retains the packaged genome until tail attachment[20]. We solved a 2.2 Å resolution structure of the stopper protein of phage 812 using X-ray crystallography (Fig. S2C and Table S6). The protein forms hexamers in solution and in the crystal (Fig. 2 and S12), which differ in structure from those in the phage 812 particle (Fig. 2A–C and S13). In the crystal structure, the six gating loops of the stopper protein form α-helices and restrict the central channel to 4.5 Å (Fig. 2A). Similarly, the central channel of the stopper complex in the tailless DNA-filled particle of phage SPP1 is blocked by gating loops in α-helical conformations (Fig. S13A–B)[8]. Therefore, we speculate that phage 812 stopper proteins use the gating loops to ensure the retention of the phage genome before the tail attachment.

## Tail attachment induces opening of the phage 812 stopper channel

The gating loops of the stopper proteins in the phage 812 virion are in β-hairpin conformations that leave the central channel open (Fig. 2B and S13C). To transform from the closed conformation observed in the crystal structure into the open form in the phage 812 virion, stopper proteins rotate 15° and pull the gating loops away from the adaptor complex (Fig. S13D). This rotation is induced by the attachment of the tail terminator complex, which pushes tetra-cysteine loops of stopper proteins outwards (Fig. S13CF). The Cys-Pro-Cys-$(X)_n$-Cys-X-X-Cys pattern of the tetra-cysteine loop is conserved in stopper proteins of phages from the family *Herelleviridae* (Fig. S13E), but not among any other phage groups. Thanks to proline and cross-strand zinc coordination (Fig. 2D), the loop is stable and moves as a rigid body together with the core β-barrel domain of the stopper protein (Table S7). The gating loop changes conformation to a β-hairpin and lines the inner surface of the neck channel, which delimits the local channel diameter to 40 Å (Fig. S13CF). The α-helix-to-β-strand transformation of the gating loop is enabled by the residues Gly106, Gly113, and Gly118, which serve as hinges and provide conformational freedom to the polypeptide chain. These glycine residues are conserved within the *Herelleviridae* family (Fig. S13E). The coupled conservation of the tetra-cysteine motif and the glycine hinges of the gating loop in stopper proteins indicates that the mechanism of genome gating is shared among *Herelleviridae* phages.

## Genome release is not dependent on changes in neck structure

We induced genome release in purified phage 812 virions by in vitro treatment with urea. The capsids showed a continuum of residual genome density, as the mechanisms that ensure completion of DNA ejection in vivo cannot occur. We selected capsids with the lowest average inner density, and we termed this dataset 'genome release intermediates' (Fig. S14). We solved the structure of the neck of the genome release intermediates using twelvefold and sixfold symmetries to resolutions of 3.1 and 3.6 Å, respectively (Fig. S14–17 and Table S1). The neck of the genome release intermediate lacks decoration proteins, which have been stripped during the tail contraction treatment, and there is no resolved density for the N- and C-termini of stopper proteins that interact with the stopper decoration proteins in the virion (Fig. 1D and S18A–B). The crown domains of portal proteins are shifted 8.9 Å away from the core of the portal complex and rotated 28° counterclockwise around the portal central axis, when looking from the head center, relative to their position in the virion (Fig. 1D, RMSD of 6.1 Å over 5,256 portal protein Cα atom pairs). This structural change is likely due to the absence of most of the packaged DNA in the genome release intermediate. Aside from the change in position, enabled by flexible linkers connecting crown and wing domains of portal proteins, minimal difference was observed when crown domain models of virion and genome release intermediate were compared. The other domains of portal proteins were not affected by tail contraction and genome release (RMSD of 1.1 Å over 4,368 Cα atom pairs). Adaptor, stopper, tail terminator, and neck-adjacent tail tube complexes of the genome release intermediate are in the same conformations as in the virion (RMSD of 0.78 Å over 6942 Cα atom pairs available for comparison, Fig. 1D). The absence of structural differences between the neck proteins of the phage 812 genome-containing particles

**Fig. 2 | Structure of stopper protein in crystal and phage 812 particle. A–B** A hexamer of stopper proteins in the crystal (**A**) and in the virion (**B**), shown in cartoon representation. Alternating sub-units are shown in color and in gray. **C** Structural alignment of stopper protein monomers from the crystal (teal) and the virion (magenta). The two positions of the gating loop are indicated with orange asterisks, those of the C-terminus with blue asterisks, and those of the tetra-cysteine loop with a golden asterisk. **D** A zinc atom (blue sphere) is coordinated between four cysteine side chains in the crystal structure of the tetra-cysteine loop. Distances are reported in angstroms.

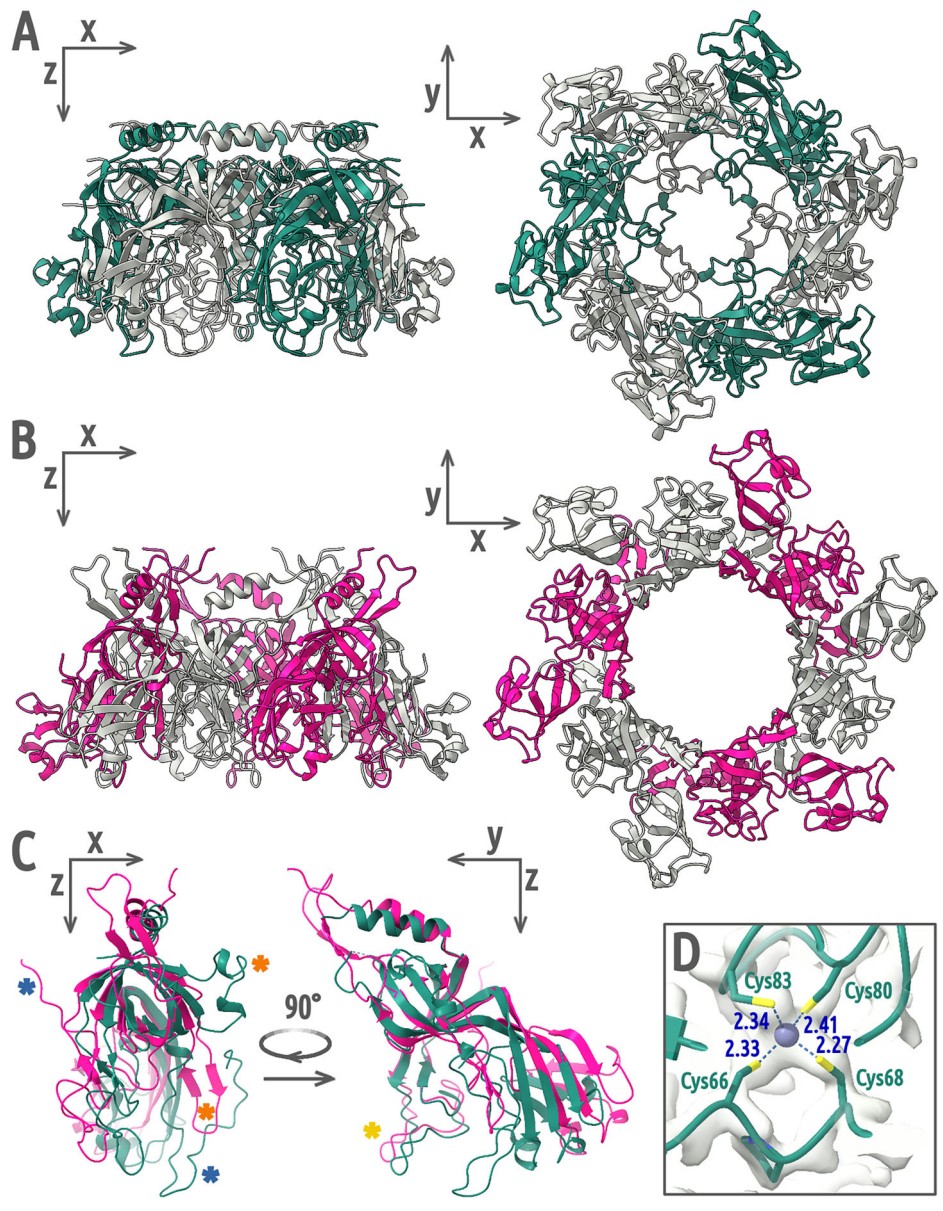

and genome release intermediates indicates that no conformational change to the neck is involved in the regulation of phage 812 genome release. Therefore, it is likely that the genome release is enabled by the release of the tail tape measure proteins, as discussed below.

**Changes in the neck-to-tail contacts caused by tail contraction**
The tail terminator complex caps the tail and interacts with the terminal hexamers of tail tube and tail sheath proteins. The core β-sheets of six tail terminator proteins build a β-barrel of the same size and shape as the tail tube β-barrel, extending the tail channel seamlessly into the neck (Fig. S6DH). The interaction is strengthened by N-termini of tail tube proteins which insert into gaps between cores and β-sandwich domains of tail terminator proteins. Tail sheath contraction and genome ejection have no effect on the conformation of the terminal tail tube complex nor on its interface with the tail terminator complex (Fig. 1D).

The interfaces between tail sheath and tail terminator proteins, and between tail sheath proteins from two consecutive discs in the tail of the phage 812 virion, are similar (Fig. 3A). Tail terminator or tail sheath proteins from a hexamer located closer to the phage head provide C-terminal β-strands to complement β-turns of domains I of tail sheath proteins from a

disc located further away from the head (Fig. 3A). Inter-disc β-sheet complementation was observed also in phages E217, Pam3, and Pa193 and appears to be a conserved feature of phages with contractile tails[9,10,25]. The tail terminator protein of phage 812 is unique in having an outer β-sandwich domain that clamps tail sheath domain I against the tail tube (Fig. 1). After tail sheath contraction of phage 812, the inter-disc contacts between tail sheath proteins are preserved, and domains I and C-terminal β-strands of tail sheath proteins move as rigid bodies to their positions in the contracted tail sheath[4]. In contrast, the tail terminator protein does not change conformation upon tail contraction, and the domain I of the interacting tail sheath protein remains locked in the native position while the other tail sheath protein domains move (Fig. 3B and S6I). Domain II moves to the canonical contracted position, while outer domains III and IV are not well resolved in the reconstruction, indicating that they become mobile (Fig. 3C). The helix connecting domain I to domain II is partially unraveled to accommodate the increased distance between the locked domain I and the shifted domain II (Fig. 3B). In comparison, following the tail contraction of phage E217, domains I of neck-adjacent tail sheath proteins move away from the gateway protein gp29, a homolog of phage 812 tail terminator protein, as gp29 lacks a β-sandwich domain locking the tail sheath domain I.

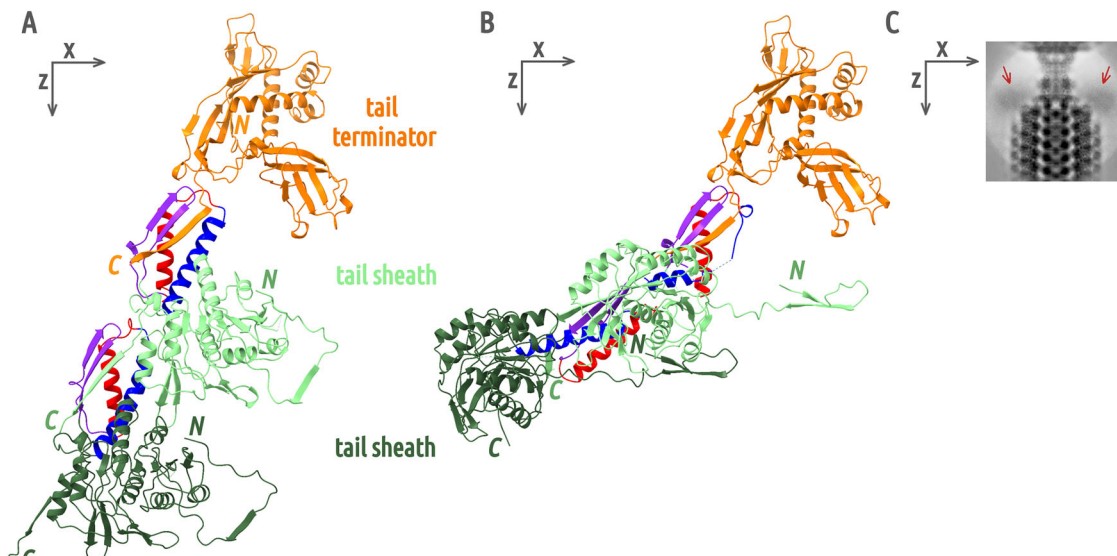

**Fig. 3 | Tail terminator and tail sheath proteins in native and contracted tail.**
**A** Comparison of intermolecular contacts between tail terminator and tail sheath proteins in the phage 812 virion. A tail terminator protein is shown as an orange cartoon, an adjacent tail sheath protein in light green (domains III and IV were omitted) and a tail sheath protein from the second disc in dark green. Domain I segments of the tail sheath proteins highlighted in blue (α-helix residues 490–513), red (α-helix residues 522–538), and violet (β-turn residues 551–563) interact with the C-terminus of the adjacent tail sheath or tail terminator protein. **B** Comparison of contacts between tail terminator and tail sheath proteins after tail contraction.

Domains III and IV of tail sheath proteins were omitted for clarity. The coloring of the proteins is identical to (**A**). The C-terminal β-strand of the tail terminator protein, and violet and red segments of domain I of the adjacent tail sheath protein, retain the same position as in the native structure. However, residues 490–513 (in blue), which in the native tail form an α-helix, reorganize. In contrast, the homologous interface between two adjacent tail sheath proteins retains the same structure as in the native tail, but its position is shifted. **C** Cryo-EM image of the neck/tail junction of the phage after tail contraction, showing the disordered state of domains III and IV of tail sheath proteins from the neck-adjacent disc (red arrows).

The C-terminal β-strand of gp29 of phage E217 swings outwards to accommodate the conformational change[10].

## Binding of the DNA to the outer surface of the portal complex

A section of dsDNA, which we named 'anchor DNA', encircles the portal complex at the level of wing domains in the cryo-EM reconstruction of the phage 812 genome release intermediate (Fig. 4A, B and S18). In an asymmetric reconstruction, this DNA is resolved as a split ring with a right-handed twist and a radius of 65 Å (Fig. 4A, B). The DNA is in the B-form. The diameter of the portal complex at the level of the DNA binding site is such that the DNA makes one helical turn per portal monomer, permitting an analysis of DNA-portal interactions in twelve and sixfold symmetrized maps (Fig. 4C–E and S18A, B). Side chains of Arg137, Arg145, and Lys425 of the portal protein form salt bridges with DNA backbone phosphates (Fig. 4C–E). These DNA-binding side chains are stabilized by contacts with negatively charged side chains of Glu139 and Glu142 (Fig. S19). Positively charged DNA-binding sites of neighboring portal proteins are separated by negatively charged valleys formed by Asp130, Asp132, Glu134, and Glu424 (Fig. S7B, C). The DNA minor groove facing the portal protein is narrowed to 3 Å, from *ca* 6 Å in straight B-form DNA (Fig. 4F)[28]. The side chain of Lys138 extends into the minor groove and interacts with two consecutive base pairs (labeled as F5:R6 and F6:R5 on strands F and R, Fig. 4EH). The nucleotide identity of these two base pairs cannot be determined based on the cryo-EM map due to rotational and inter-particle averaging. However, molecular simulations with all nucleotide combinations of these two base pairs (Table S8) show that a strong base pair (containing G and C) is disfavored in both positions due to the presence of the guanine N2 hydrogen donor clashing with the Lys138 NZ hydrogen donor (Table S9). This belt of charged side chains on the wing domains of phage 812 portal complex therefore provides a high-avidity, shape-specific, and sequence-sensitive binding site for B-form DNA.

To determine the sequence of the genome interacting with the portal complex, we sequenced the DNA in phage 812 virions and genome release intermediates. A long terminal repeat (LTR) of 8.5 kbp flanks both

chromosomal ends of the phage 812 genome packaged in virions[2]. The repeats are conventionally named as left, the repeat at the 5' end of the genomic sequence, and right, the 3' end repeat downstream of the genomic sequence[29]. The vast majority of sequence reads from the genome release intermediate start at position 1 of the LTR (Fig. 4G and S18C), indicating that the DNA retained in the capsid corresponds to the left end of the phage 812 genome. This sequence is likely to be the sequence bound to the portal complex (Fig. 4A–B). Our results also demonstrate that phage 812 genome ejection proceeds from the right chromosome end toward the left end, which corresponds to a first in, last out direction.

In the asymmetric cryo-EM reconstruction of the phage 812 genome release intermediate, the ends of the anchor DNA split ring are positioned next to each other (Fig. 4A–B). One end of the DNA points toward the portal-capsid junction, and the other end points to the top of the portal crown. The crown funnel, connected to the core of the portal complex by flexible loops, is tilted 3.5° off the neck axis towards the opening in the anchor DNA ring (Fig. S20), indicating that the tilting is caused by the interactions of the DNA loop with the portal crown. Furthermore, the twelve crown termini of portal proteins fold into six structures reminiscent of molten globules (Fig. S18D–F) and contain 41% of residues with charged side chains, making their interaction with DNA plausible.

The interaction of the genome with the portal complex is described here for the phage 812 genome release intermediate. We speculate, however, that the primary purpose of the molten-globule portal crown structures and of the B-DNA-binding circular belt formed by wing domains is to anchor the leading end of the genome inserted into the capsid during the initial stage of genome packaging. Based on the evidence that the genome ejection proceeds in the first in, last out direction, the interaction of the end of the DNA with the portal would be established at the beginning of genome packaging and retained until its ejection. DNA anchoring may prevent its accidental escape from the capsid caused by slippage of the packaging terminase[16]. Furthermore, the DNA anchoring also constrains the movement of the genome inside the capsid, which may prevent its knotting[30], and initiate coaxial genome spooling (Fig. S11B).

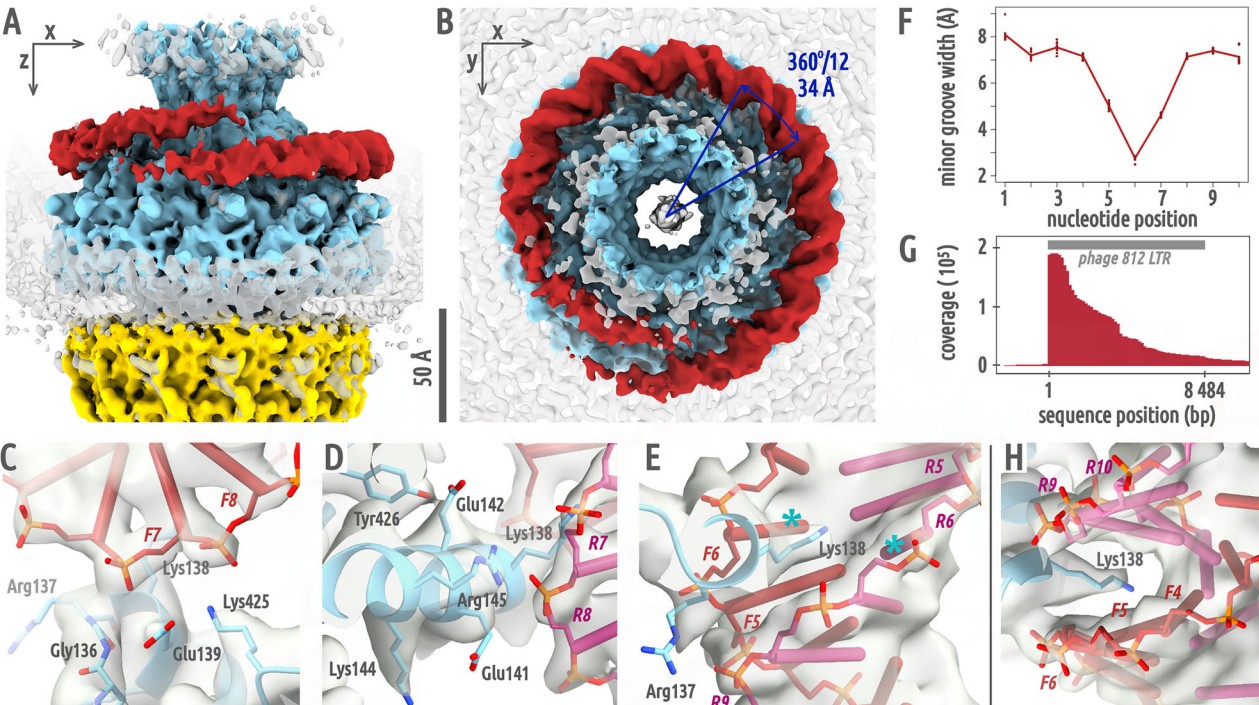

**Fig. 4 | DNA forms split ring encircling portal complex of phage 812 genome release intermediate. A, B** Surface representation of an asymmetric cryo-EM reconstruction of the neck of phage 812 genome release intermediate (EMD-18395), low-pass filtered to 6 Å and zone-colored based on its structural components: dsDNA in red, portal proteins in light blue, adaptor proteins in gold; remaining density in semi-transparent gray. The arc distance between neighboring portal wing domains described by the anchor DNA is 34 Å (indicated in navy blue), which corresponds to one B-DNA helix turn. **C–E** Interactions between side chains of portal proteins and DNA in C12-symmetrized reconstruction of the neck of the genome release intermediate (EMD-18213). The two DNA strands are colored in primary red (strand F) and rose red (strand R). Turquoise asterisks denote the bases forming putative hydrogen bonds with the Lys138 side chain. **F** Plot of the average van der Waals width of the minor groove over one helical turn of the DNA anchored to the portal complex (n = 12 helical turns). **G** Mapping of sequence reads of DNA remaining in phage 812 capsids after genome release onto phage 812 long terminal repeat (LTR). **H** Shape of the DNA minor groove facing the portal protein binding site.

## Interaction of the portal complex with DNA inside the neck channel

In the phage 812 virion, DNA fills the neck channel and is in contact with tunnel loops of the portal proteins (Fig. 5A). Each tunnel loop interacts with the crown base of the same portal protein subunit, which pushes the loop against channel DNA (Fig. 5A). The neck channel of the genome release intermediate contains residual DNA (Fig. 5B). The crown domains of portal proteins of the genome release intermediate are raised relative to their positions in the virion, which releases the pressure on the tunnel loops that ease away from DNA (Fig. 5B). Nevertheless, an asymmetric reconstruction of the portal channel of the genome release intermediate shows that the tunnel loops remain in contact with DNA (Fig. 5C). Tunnel loops (residues 376–381) are stabilized by interactions with the backbone of the DNA (Fig. 5C–D). The side chain of Lys381 extends towards DNA phosphates (Fig. 5E). We speculate that the lowering of the crown caused by the pressure exerted by the encapsidated DNA leads to a compression of the portal tunnel loops and tighter contact with the channel DNA (Fig. 5A–B). It was hypothesized that this conformational switch sends the headful packaging termination signal through DNA to the terminase of phage SPP1[17], and a similar mechanism could also trigger packaging termination in phage 812.

## B-to-A DNA transition inside the neck channel could regulate genome translocation

In the asymmetric reconstruction of the neck of the genome release intermediate, the DNA leans away from the central axis at the portal/adaptor junction (Fig. 6A, B). The bent DNA forms asymmetric contacts with side chains of Lys334 from portal clip domains (Fig. 6A, B). Rings of lysines lining the inside of the portal channel at the clip domain level were previously identified in many tailed phages and were speculated to interact with the channel DNA[6,31,32]. Inside the adaptor chamber, the DNA switches from the B to the A-form over one helical turn before resuming the B-form upon entering the β-hairpin tube of the adaptor complex (Fig. 6A). A-form DNA is characterized by a wider diameter compared to the B-form, an off-center displacement of nucleic base pairs, and an exposed minor groove (Fig. S21)[28]. The A-form region of the DNA interacts asymmetrically *via* its minor groove with short loops (residues 265–272) of adaptor proteins at the exit from the adaptor chamber, where side chains of Met269 and Tyr270 are positioned to form hydrophobic interactions with DNA ribose sugars (Fig. 6AC). In the narrow β-hairpin tube of the adaptor complex, with a van der Waals diameter of <24 Å, the B-DNA is recentered on the channel axis by contacts between DNA backbone phosphates and Glu254 side chains (Fig. 6AD). The transition from an off- to on-axis position is enabled by a bend in the DNA double-strand due to a sharp narrowing of its minor groove (Fig. 6E, F and S22).

There is a similar widening and off-centricity of the channel DNA in the adaptor chamber of the phage 812 virion as in the genome release intermediate (Fig. S23). The B-to-A conversion must be therefore reversible and could occur when genome progression through the neck is slowed or paused. B-to-A transitions are typically induced by dehydration or by DNA-remodeling proteins and are sequence-dependent[33]. We hypothesize that in phage 812, the double strand bends inside the adaptor β-hairpin tube (Fig. 6E, F and S22A), is pressed against hydrophobic loops of the adaptor proteins (Fig. 6C), undergoes a flipping of ribose sugars to the exposed 3'-endo conformation, and so triggers DNA conversion to the A-form[34]. AT-rich dsDNA is prone to bending under the influence of positive ions[35–37], and GC-rich dsDNA enables B-to-A form conversion[33,38]. Therefore, an AT-rich DNA tract located in the adaptor β-hairpin tube followed by a GC-rich tract located in the adaptor chamber is susceptible to forming the structure

**Fig. 5 | Cross-sections through cryo-EM maps of phage 812 portal channel. A, B** Twelvefold-symmetrized reconstructions of the portal in the virion (A, EMD-18445) and genome release intermediate (B, EMD-18213) are shown as gray semi-transparent surface. Portal proteins are shown in light blue in cartoon representation. Blue arrows indicate the position of tunnel loops, and red asterisks denote DNA rings around the portal complex. **C–E** Asymmetric reconstruction of the portal channel of the phage 812 genome release intermediate (EMD-18372) viewed perpendicular to (**C**) and along (**D, E**) the tail axis. The portal complex is in light-blue cartoon representation, the adaptor complex in gold, and the DNA in red. The tunnel loops of portal proteins (residues 375–392) extend towards the DNA. The contacts with the DNA phosphate backbone are mediated by side chains of Lys381 (**D**) and the flexible loop Gly376–Gly379 (**E**), which are indicated by blue arrows.

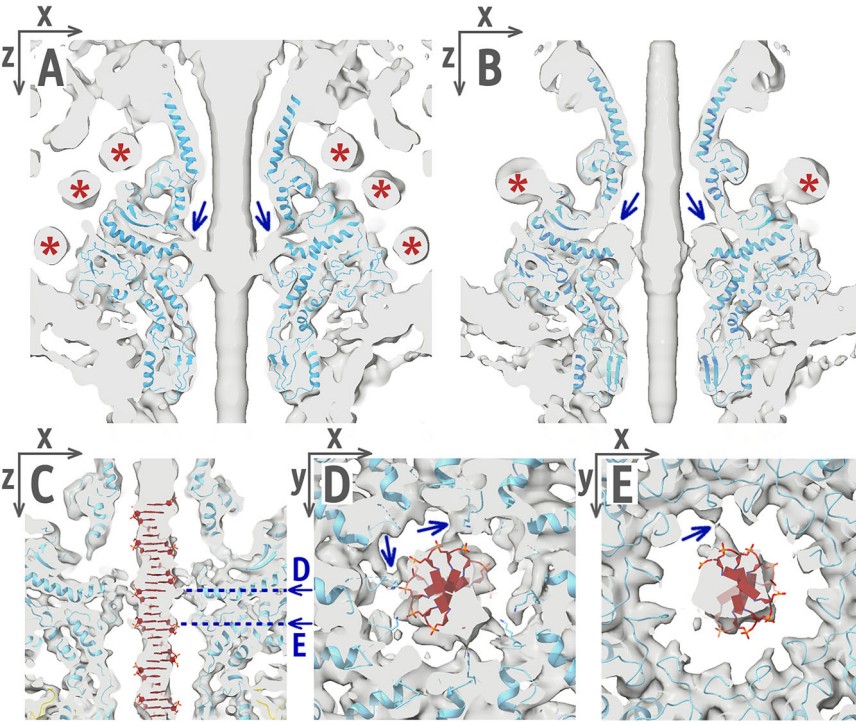

observed in phage 812 adaptor channel. This DNA remodeling could ease the pressure of the DNA packaged inside the virion capsid on tail tape measure proteins in the tail tube, as the off-axis A-DNA region likely transfers part of this pressure to the adaptor complex. A similar arrangement of dsDNA inside the neck was observed in phage φ29 virion, in which the genomic DNA appears to adopt a toroidal shape inside the chamber of the lower collar complex, while the DNA terminus occupies the lower collar tube and is in contact with the tail tape measure protein-analog gp3[39,40]. During genome delivery, the A-form conversion in the adaptor chamber of phage 812 could provide a means to strategically pause DNA ejection that can be encoded into the genomic sequence through modulating the AT/GC composition of the translocating DNA. Indeed, the sequencing of the residual capsid DNA of phage 812 after the induction of genome ejection shows an accumulation of reads of certain lengths (Fig. S24), consistent with genome ejection pausing. The pausing may contribute to the regulation of phage gene expression in the initial stages of infection, help to avoid targeting by bacterial immune systems before phage counter-proteins are synthesized, and prevent phage DNA degradation during phage-induced cleavage of the host genome, as observed in phage T5[41,42]. Pull on the stalled DNA caused by its diffusion in the bacterial cytoplasm or by the activity of RNA polymerase could resume the genome ejection[43,44].

### Genomic end and the tail tape measure protein

The van der Waals diameter of the channel formed by the stopper, tail terminator, and tail tube proteins is >30 Å, and the walls are negatively charged, allowing an unhindered passage of DNA (Fig. 7 and S25). In the asymmetric reconstruction of the neck/tail junction of the phage 812 virion, the four neck-proximal rings of the tail tube are occupied by dsDNA (Fig. 7A). In the fifth ring of the tail tube, the end of the DNA double-strand is in head-to-head contact with a trimeric density that we attribute to the N-termini of the tail tape measure proteins (TMP) (Fig. 7A). Thus, the packaged genome is poised for ejection upon the release of the TMP complex from the tail tube. Unlike phage 80α[21], phage 812 does not contain a tail completion protein blocking DNA inside the neck channel, pointing at different genome gating mechanisms used by these two phages.

The length of the channel from the adaptor complex to the fourth tail tube ring in the phage 812 virion is 269 Å, which corresponds to a DNA

length of *ca* 85 bp (Fig. 7)[28]. The DNA has a pitch of *ca* 35 Å/turn, typical for B-form DNA[28], but a shallow minor groove caused by an off-axis displacement of the bases (Fig. 7B), which is a mark of A-form DNA[28]. The 85 terminal base pairs of the right end of phage 812 long terminal repeat contain four poly(G):poly(C) tracts, which could facilitate the formation of the hybrid B/A-form (Fig. 7C)[38]. Phages from the *Twortvirinae* subfamily of *Herelleviridae* have genomes with an AT content above 70%, therefore the presence of clustered G-tracts at the end of phage 812 genome is particularly notable. We hypothesize that the hybrid DNA form is caused by the pressure exerted by the packaged genome since the A-form of DNA accommodates 30% more base pairs per axial length than the B-form[28]. A complete conversion of the DNA end to the wider A-form is prevented by the limited 31 Å van der Waals diameter and the negative surface charge of the tail tube channel (Fig. S25). In the structure of the phage λ virion, the genomic end is likewise located inside the tail channel and has a hybrid B/A conformation with a shallow minor groove[22]. This DNA remodeling could serve to dissipate the energy of impact exerted by the DNA end on the tail tape measure protein when the DNA descends into the tail tube after head-to-tail joining, analogous to suspension coils in a car.

### DNA compresses the tail tape measure proteins

The TMP complex acts as a scaffold for tail tube assembly and determines its length[45–48]. However, in the phage 812 virion, the tail tube exceeds the length of the TMP trimer by four rings of tail tube proteins (Fig. 7). This length difference corresponds to 166 Å, which is 8% of the total tail tube length, or *ca* 110 amino acid residues arranged in an α-helix. Mass spectrometry data indicate that both N- and C-termini of TMP are present in phage 812 virions (Fig. S26). Therefore, we hypothesize that the TMP trimer inside the tail tube of the phage 812 virion is longitudinally compressed by the DNA released from the neck after head-to-tail joining. Based on the secondary structure prediction of phage 812 TMP (Fig. S26A) and known structures of its homologs[24,49], we speculate that the TMPs form a trimer with parallel α-helical coiled-coil segments interspersed with segments without a specific secondary structure. The segments without predicted secondary structure account for 42% of the total sequence, and, unlike coiled coils, can be compressed. TMP compression could disrupt its interactions with tail tube proteins and facilitate TMP release prior to genome ejection.

**Fig. 6 | DNA converts to A-form in adaptor channel. A–D** An asymmetric reconstruction of the adaptor channel of the genome release intermediate (EMD-18372) is shown as gray semi-transparent surface. Portal proteins are shown in cartoon representation in light blue, adaptor proteins in gold, stopper proteins in magenta, and DNA in red. The position of the central channel *z*-axis is indicated by a dashed gray line. Side chains of Lys334 of portal proteins and Glu254, Met269, and Tyr270 of adaptor proteins are shown in stick representation. The numbering of nucleotides for the measurement of the minor groove width in panel (**E**) is indicated in panel (**A**) on the strand oriented in the head-to-tail direction. Nucleotides 1–10 are in the A-form. **E** Plot of the Van der Waals width of the DNA minor groove. The dashed turquoise line represents the minor groove width of an ideal B-form DNA. **F** Cross-section through the channel density tilted -24° along *x*, showing the narrow minor groove of DNA near the Glu254 ring.

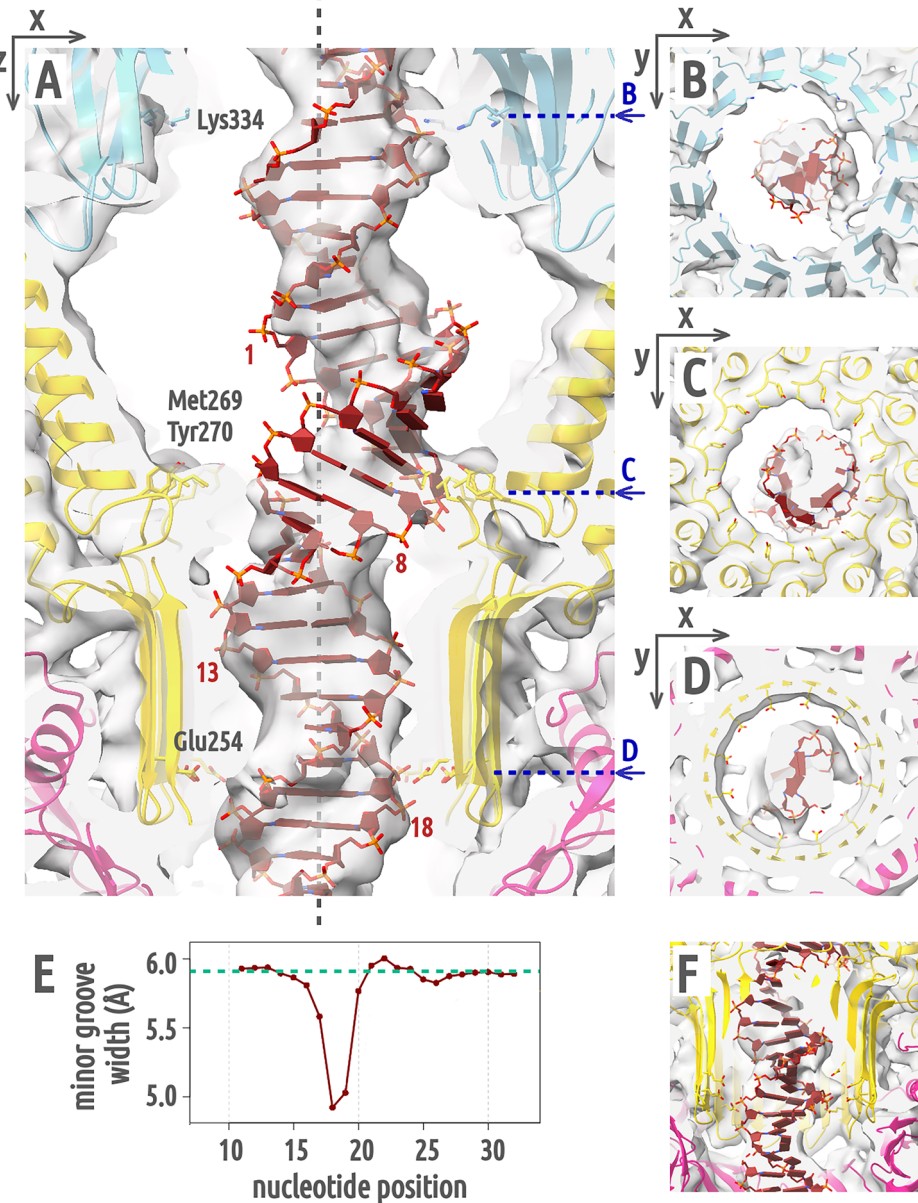

## Conclusions

The comparison of the structures of phage 812 virion and contracted particle highlights the mechanistic roles that the neck proteins and the DNA play in the phage life cycle, from genome packaging through virion assembly to genome ejection (Fig. 8). The portal provides a circular, high-avidity binding site that anchors the leading end of the phage genome and thus influences its spooling inside the head. Pressure exerted by the packaged genome induces a lowering of the crown of the portal complex. The B-to-A-form DNA remodeling by side chains of the portal and adaptor complexes stabilize the genome in the virion and could strategically pause its translocation during ejection. The stopper proteins in the closed conformation block the central channel with their gating loops, suggesting a mechanism for securing the genome inside tailless heads. Conformational changes to the gating loops, induced by the attachment of the tail terminator complex during virion assembly, open the stopper channel and permit the progression of the genomic end into the tail tube. The hybrid B/A-form adopted by terminal G-tracts of the phage DNA and the compression of the tail tape measure protein trimer likely serve to disperse the force exerted by the fully packaged genome. Contrary to expectations, the neck proteins of phage 812 do not play a direct role in regulating the genome ejection from the capsid, as the tail sheath contraction does not affect their structure. Instead, disruption of the interactions of TMPs with the baseplate, induced by baseplate binding to a host cell[4], likely enables expulsion of the TMPs and genome ejection by unblocking the tail tube channel.

## Methods

### Preparation of phage samples

*Staphylococcus* phage 812, strain K1/420, was propagated in *S. aureus* (strain CCM 4028, Czech Collection of Microorganisms) planktonic cultures grown at 37 °C in meat peptone broth supplemented with $CaCl_2$ to 2 mM final concentration[4]. The phage lysate was centrifuged at 5000 × g and 4 °C for 30 min then filtered through 0.45 μm membrane to remove cell debris. Phages were pelleted by centrifugation at 54,000 × g for 2.5 h, then gently dissolved in 500 μL of phage buffer (50 mM Tris pH 8, 10 mM NaCl, 10 mM $CaCl_2$). The sample was purified on a CsCl step density gradient (1.45, 1.5 and 1.7 g/mL of CsCl in phage buffer) by centrifugation at 194,000 × g for 4 h. Collected fractions containing phage particles were dialyzed against the phage buffer over-night. For the preparation of genome release intermediates with contracted tails used for cryo-EM data collection, the phage sample was supplemented with lipoteichoic acid (LTA) from *S. aureus*

**Fig. 7 | Structure of terminus of phage DNA and tail tape measure protein in virion tail. A** An asymmetric reconstruction of the neck-to-tail junction (EMD-18516) is shown as gray semi-transparent surface. Adaptor proteins are shown in cartoon representation in gold, stopper proteins in magenta, tail terminator proteins in orange, stopper decoration proteins in forest green, terminator decoration proteins in steel blue, tail tube proteins in violet, and tail sheath proteins in light green. An ideal B-form dsDNA (in red) was placed into the helical density inside the tail tube channel. The first 14 residues of three TMPs (in turquoise) were fitted into the channel density starting at the fifth ring of the tail tube. **B** Comparison of reconstructed DNA densities in phage 812 tail channel of the virion and neck channel of the genome release intermediate with a B-form DNA model. The neck channel DNA has clear major (orange arrow) and minor (blue arrow) groove valleys typical for B-form DNA. The tail channel DNA has a deep major groove and a shallow minor groove caused by an off-center displacement of the base-pair densities, indicating a hybrid B/A form. **C** The reverse complement (RC) sequence (85 right-end base pairs) of the phage 812 long terminal repeat is aligned with the DNA density in panel A from the adaptor β-hairpin tube to the end of the genome.

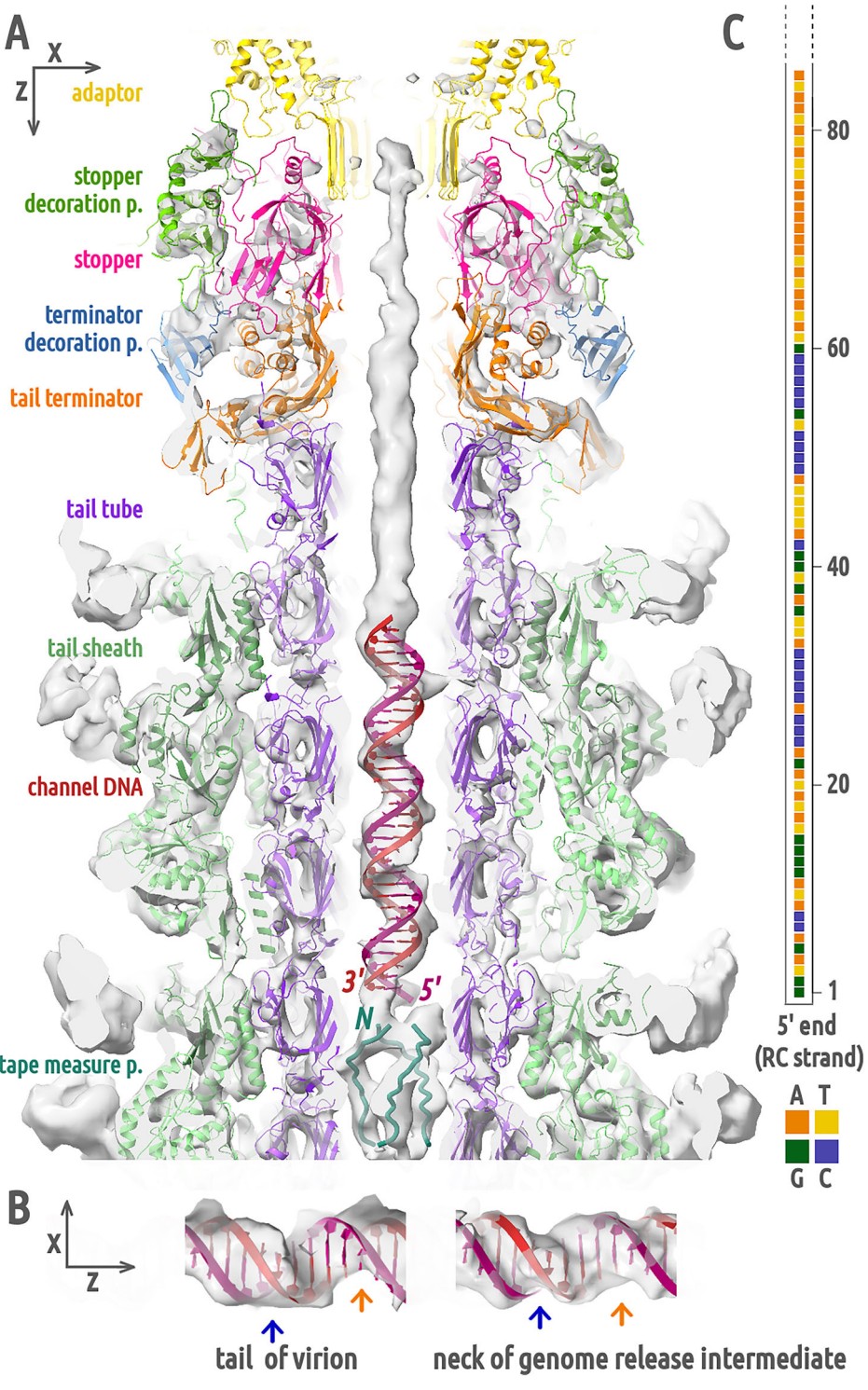

(SIGMA) to a final concentration of 100 µg/mL, then diluted 1:10 using the phage buffer with the addition of urea at a final concentration of 3 M, and incubated for 2 h at 42 °C. The sample was then diluted 1:3 with the phage buffer, incubated for 15 min at room temperature with Turbonuclease (Abnova) at a final concentration of 30 µg/mL, and pelleted for 1 h at 75,000 × g. The pellet was resuspended in 100 µL of the phage buffer. For the preparation of particles with contracted tails used for residual genome sequencing, a second phage sample was treated with a tail contraction protocol modified as follows: the dilution ratio to urea-supplemented phage buffer was 1:20, the incubation was carried out for 2 h at 42 °C then 1 h at 4 °C, and LTA was omitted.

**Mass spectrometry analysis of phage samples**

To establish the identity of protein components, a virion sample treated with Turbonuclease was analyzed by 1D tricine 15% SDS-PAGE. Selected gel bands were excised and after destaining and washing, each gel area was incubated with trypsin. Liquid chromatography–tandem mass spectrometry (LC–MS/MS) analysis was performed using UltiMate 3000 RSLC system (Thermo Fisher Scientific) on-line coupled with Impact II Ultra-High Resolution Qq-Time-Of-Flight mass spectrometer (Bruker). Exported MS/MS spectra were analyzed in Proteome Discoverer 1.4 (Thermo Fisher Scientific) using in-house Mascot 2.6.2 (Matrixscience) search engine. Mascot MS/MS ion searches were done against *Staphylococcus* phage 812

**Fig. 8 | Functions of neck proteins in phage 812 genome packaging and ejection.** Phage components are shown schematically, with capsid in dark gray, terminase in beige, portal in light blue, adaptor in gold, stopper in magenta, tail terminator in orange, tail tube in violet, tail sheath in light green, TMP in turquoise, and DNA in red. **A** Left DNA end inserted into the capsid interacts with the crown of the portal complex. **B–C** DNA binds to and loops around the wing domains of portal proteins, which directs coaxial spooling of the packaged genome. **D** The packaged genome compresses the portal crown, which could contribute to the termination of packaging and replacement of the terminase complex with adaptor and stopper complexes. Stopper gating loops in the closed conformation ensure genome retention. **E** When the tail binds to the neck, the stopper gating loops open and the genomic end descends into the tail tube. TMP and the A-form DNA inside the adaptor channel prevent the genome from further progress. **F** During genome ejection, a sequence-induced transition to A-form DNA inside the adaptor channel can pause the progress of the genome release. **G** In the last stage of genome ejection, the portal crown is lifted and the residual DNA from the left LTR remains anchored to the portal protein wing domains.

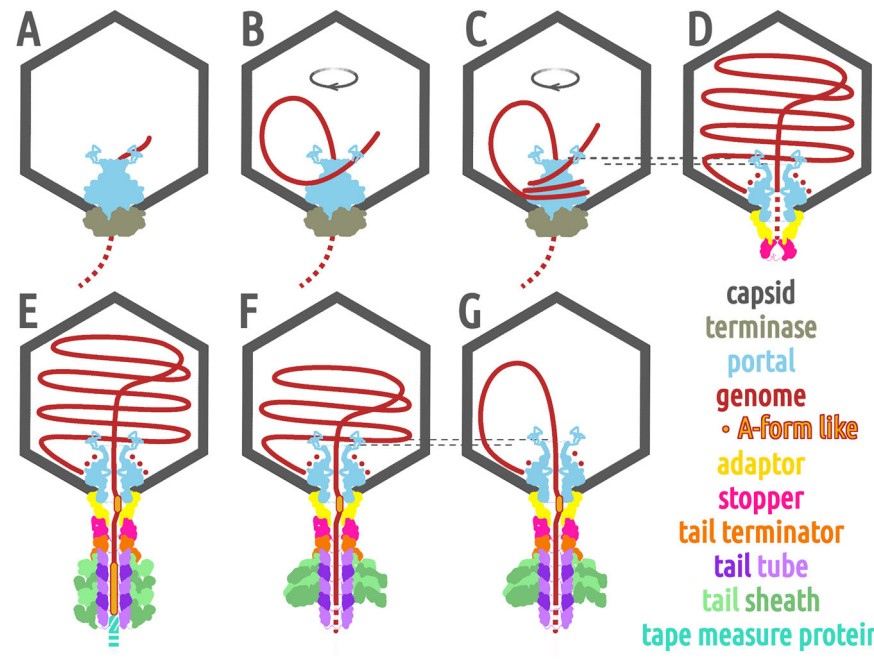

proteome (UniProt ID UP000203789) and GPM cRAP contaminant database (http://www.thegpm.org/crap/, version 2018-11-22).

### Resequencing of phage 812 K1/420 genome

Considering the limitations of the 454-sequencing method used for the previous determination of the phage 812 K1/420 genome (GenBank acc. no. KJ206563.1), a hybrid approach with both short and long reads was used to determine the precise full-length genome sequence as packaged in the capsid.

The phage lysate was pelleted by high-speed centrifugation at 54,000 × g and resuspended in 400 µL of the phage buffer. One µL of DNase I (1U/µL; Thermo Scientific) and 1 µL of RNase A (10 mg/mL, New England BioLabs) were added, and the solution was incubated for 30 min at 37 °C. Next, 30 µL of 10% SDS and 10 µL of proteinase K (20 mg/mL; Sigma-Aldrich) were added, and the sample was incubated for 15 min at 55 °C. DNA was purified using Genomic DNA Clean & Concentrator-10 (ZymoResearch) according to the manufacturer's instructions for genomic DNA isolation (the addition of 2 volumes of ChIP DNA Binding Buffer to the DNA sample).

For Oxford Nanopore sequencing, the library was prepared using an SQK-LSK114 kit (Oxford Nanopore Technologies, UK) according to the manufacturer's instructions, using Long Fragment Buffer (LFB) in the washing step. The library was sequenced with a FLO-FLG114 flow cell in a MinION device (Oxford Nanopore Technologies). The device was controlled with the software MinKNOW v23.11.4 (Oxford Nanopore Technologies), which was also used for base-calling (Super-accurate model) and trimming.

For Illumina sequencing, a 500 bp sequencing library was prepared with an NEBNext Ultra II DNA library prep kit for Illumina (New England BioLabs). The samples were sequenced using a MID output cartridge in 150 bp paired-end mode on an Illumina NextSeq sequencing platform (Illumina, San Diego, CA, USA). The quality of sequencing reads was analyzed with FastQC v0.11.8[50]. Bases of lower quality (average quality below 20) and adaptors were trimmed using the sliding window model in Trimmomatic Galaxy v0.39[51].

The complete genome was obtained using hybrid assembly in Unicycler Galaxy v0.5.0[52]. Long terminal repeats were determined based on read coverage levels[2], and confirmed by long-read mapping in Geneious Prime v2025.0.3 using the Minimap2 v2.24 plugin (https://www.geneious.com). Genes were annotated using RAST[53], with manual inspection using NCBI BLAST[54], InterPro[55], and based on the results of this study (Table S2).

Independently, long-read sequencing was used to identify residual DNA in phage particles after genome ejection. The DNA isolation procedure was the same as above, using 400 µL of purified contracted viral particles in the phage buffer and the protocol of the Genomic DNA Clean & Concentrator-10 kit (ZymoResearch) for the isolation of PCR products/DNA fragments (addition of 5 volumes of ChIP DNA Binding Buffer to the DNA sample). The sequencing library was prepared as described above, using Short Fragment Buffer (SFB) in the washing step. The sequencing was performed on an Oxford Nanopore platform as described above. For subsequent analysis, remaining Oxford Nanopore adaptor sequences (Top strand: 5'-TTTTTTTTTCCTGTACTTCGTTCAGTTACGTATTGCT-3'; Bottom strand: 5'-GCAATACGTAACTGAACGAAGTACAGG-3') were trimmed from sequencing reads using the BBDuk plug-in v38.84 (Geneious Prime v2025.0.3) with the following parameters: Kmer Length = 15, Maximum Substitutions = 1; Maximum Substitutions + INDELS = 1; Trim partial adaptors from ends with kmer length = 2. The trimmed reads with a maximum length corresponding to the LTR (8,484 bp, 93% of all reads) were mapped to the reference genome 812 K1/420 (KJ206563.2) using the Geneious mapper (Geneious Prime v2025.0.3) with the following parameters: Map multiple best matches: Randomly; Maximum Gaps Per Read: 15%; Maximum Gap Size: 50; Minimum Overlap Identity: 90%; Word Length: 12.

### Cryo-electron microscopy

The phage sample (3.5–4.0 µL) was applied onto glow-discharged holey carbon-coated copper grids (Quantifoil R2/1, 200 or 300 mesh) and vitrified in liquid ethane using a Vitrobot Mark IV (Thermo Fisher Scientific). Cryo-EM data were collected on a Krios TEM (Thermo Fisher Scientific) operating at 300 kV. The beam was aligned for parallel illumination in NanoProbe mode, and coma-free alignments were performed to remove the residual beam tilt. The virion sample was acquired using a Falcon II detector (Thermo Fisher Scientific) in linear mode with 1 s exposure at a magnification of 75,000 ×, resulting in a pixel size of 1.061 Å. Genome release

intermediates with contracted tails were collected with SerialEM 3.7.14[56] using a K2 Summit detector (Gatan) equipped with an energy filter and the slit set to 20 eV. The data were collected with 7 s exposure at a magnification of 130,000 × in super-resolution counting mode, resulting in a non-super-resolution pixel size of 1.057 Å. Complete details of the acquisition process are listed in the Supplementary Information (Table S1).

## Cryo-EM data processing

Dose-fractionated movies were aligned and dose-weighted using Motion-Cor2 1.4.0[57], and defocus estimation was performed using Gctf 1.0.6[58]. For phage virions, capsids were manually picked with EMAN2 *e2boxer*[59]. Capsids of genome release intermediates with contracted tails were automatically picked with Gautomatch 0.56[60] and virion-like capsids containing genome were removed by 2D classification in Relion 3.1[61]. Further removal of partially full capsids was performed by radially averaging capsid images using xmipp 3.0[62] and discarding images with the highest average inner capsid density calculated using R 4.1.2[63]. 3D refinement with imposed icosahedral symmetry was done with the Relion auto-refine procedure using phage 812 virion capsid (EMD-8304)[4] as an initial model. Neck vertices were identified by expanding the icosahedral symmetry of aligned capsids and subjecting them to 2D classifications in Relion with fixed orientations while applying a soft neck mask on the capsid vertex corresponding to the null in-plane angle. Neck sub-particles were extracted using a modified local extraction protocol[64] and refined applying C6 or C12 symmetries using a modified version of Relion that allows for a free rotational search while restricting the orientational search of tilt and psi angles. The post-processing of refined maps was performed in Relion. The pixel size of the dataset acquired on the Falcon II detector was adjusted to 1.080 Å during the post-processing step. Map resolution was estimated by Fourier shell correlation using the 0.143 threshold criterion. Local resolution estimation was performed by sampling the volume using a soft spherical mask 12.5 Å in diameter, then the map was low-pass filtered and B-factor sharpened during post-processing according to local resolution estimates. Asymmetric and symmetry-reducing reconstructions were achieved by symmetry-expanding the pre-aligned particle dataset and performing focused 3D classifications without orientational searches. For the reconstruction of the portal crown and the anchor DNA in genome release intermediates, the density of the capsid was subtracted from the particle images prior to sub-particle extraction. To compare reconstructions of portal tunnel loops in the virion and genome release intermediate, we performed automated B-factor sharpening[65] and detector MTF correction on each map in Relion, then applied a ΔB-factor of 179 Å² and a low-pass filter with a 6 Å cut-off to the maps. Detailed workflow charts of the reconstruction steps are provided in the Supplementary Information (Fig. S1 and S14).

## Model building and refinement in cryo-EM density

Protein main chains were traced in the reconstructed maps in Coot 0.9[66] and identified by comparison with structure predictions based on the primary sequence generated by RaptorX[67] or AlphaFold2[68]. The highest-ranking prediction models (for AlphaFold2 after force-field relaxation) were fitted into the maps and rebuilt in Coot. The models were refined iteratively using ISOLDE 1.4[69] and the Phenix 1.19[70] real space refinement protocol, and their quality was assessed by MolProbity 4.5.2[71] and the Phenix validation protocol (Tables S1 and S4). A circular pseudo-model of the anchor dsDNA was created in Coot, by generating a 10 bp random-sequence B-form DNA, performing restrained flexible fitting into the cryo-EM density, and extending the model 12-fold. The geometry of the DNA model was refined in the C12-symmetrized map in the presence of the portal complex using Phenix and validated using DNAtco 3.2[72]. The circular model was extended by 19 bp, then flexibly fitted to the asymmetric split-ring density in Coot. A random-sequence 63 bp-long pseudo-model of the neck channel dsDNA was generated using Web 3DNA 2.0[73] as an 11 bp-long A-form DNA flanked by B-form DNA regions, flexibly fitted to the asymmetric density in Coot, refined in Phenix in the presence of portal and adaptor dodecamers without applying symmetry operators, and validated by DNAtco. Simulated

electron densities of ideal A- and B-form DNA were generated using xmipp from atomic models created in Coot.

## Preparation of recombinant stopper protein gp97

The isolation of phage genomic DNA was performed by phenol/chloroform/isoamyl extraction[2,74]. The coding sequence for gp97 of phage 812 K1/420 was PCR-amplified from the phage DNA and inserted using ligation-independent cloning into a pET22T vector carrying N-terminal hexahistidine and SUMO Smt3 tags under the control of a T7 promoter, then introduced into an *E. coli* BL21(DE3) cell line (Novagen). The cells were grown in 1 L of autoinduction medium (ZYP-5052)[75] at 30 °C and 250 RPM until proliferation arrest, harvested by centrifugation (20 min at 3,000 × g), resuspended in 50 mM Tris pH 8.0, 150 mM NaCl, 1 mg/mL lysozyme, 0.5% n-octyl β-D-thioglucopyranoside, 125 U/mL Turbonuclease (Abnova), and lysed using EmulsiFlex-C3 (Avestin). The lysate was clarified by centrifugation (20 min at 21,000 × g, 10 °C) and filtration through a 0.45 μm membrane. The sample was then subjected to nickel affinity chromatography in a HisTrap HP column (GE Healthcare) equilibrated with 50 mM Tris pH 8.0, 150 mM NaCl, 20 mM imidazole buffer, and eluted with 350 mM imidazole buffer. The eluted fractions were buffer exchanged into 50 mM Tris pH 8.0, 100 mM NaCl in a HiPrep 26/10 Desalting column (GE Healthcare). After the cleavage of N-terminal tags by in-house produced Ulp1ac protease (at 4 °C overnight), the sample was loaded into a HisTrap column to remove the cleaved His$_6$-Smt3 tag. The untagged protein in the flowthrough was concentrated in Amicon® Ultra-15 centrifugal filter units (Merck) with a 10 kDa cut-off, and subjected to size exclusion chromatography (HiLoad 16/600 Superdex 200 pg column, GE Healthcare, calibrated using Cytiva LMW and HMW Gel Filtration Calibration kits). Fractions containing homogenous gp97 were pooled, concentrated to 3.3 mg/mL, and used for crystallization using the hanging drop vapor diffusion method by mixing the sample with 1.6 M NaCl, 8% PEG 6000, 20% glycerol, and 3% dextran sulfate sodium salt in a 1:2 ratio. The identity of the purified protein was verified by mass spectrometry prior to crystallization.

## X-ray data collection, processing, and model building

The crystals were flash frozen in liquid nitrogen without additional cryoprotectant and used to collect X-ray diffraction data at 77 K at the PROXIMA-1 beamline at the Soleil synchrotron (France) using the Dectris Pilatus 6 M detector at 12.67 keV with an oscillation angle of 0.1°. The data were indexed and processed to a resolution of 2.2 Å in space group P6$_1$2$_1$2$_1$ using the XDS software package (version 2016-11-01)[76] and scaled and merged with CCP4 Suite 6.4 Aimless[77]. Further processing was carried out in CCP4 Suite 8.0[78]: phasing was performed in Phaser 2.8.3[79] using molecular replacement with segmented subdomains from an AlphaFold2 gp97 structural model, and the model was iteratively rebuilt in Coot 0.9[66] and refined in reciprocal space in RefMac5 5.8[80], and in real space in ISOLDE 1.4[69]. The data and model validation was carried out with Phenix 1.19 Xtriage[81] and MolProbity 4.5.2[71]. The hexameric state of gp97 was validated using PDBePISA 1.48[82], and the structural features of the metal-coordinating tetra-cysteine site were assessed using the CheckMyMetal tool[83]. Further details about data processing and validation are listed in the Supplementary Information (Table S6).

## Bioinformatic analysis

A structural comparison of neck proteins against the full PDB database was carried out using the Dali server[84]. A structural homology query of the stopper gp97 tetra-cysteine motif was performed with HHpred[85] against the PDB70 database. A sequence similarity search with gp97 against non-redundant sequences of *Caudoviricetes* was done with NCBI PSI-BLAST[54] using the BLOSUM62 scoring matrix and default parameters. Representative species of *Herelleviridae* genera (Table S10) were selected according to the International Committee on Taxonomy of Viruses classification, a multiple sequence alignment of their stopper proteins was generated with Cobalt[86], and the consensus motifs were visualized as sequence logos[87].

**Article**

Similarly, to analyze the conservation of positively charged residues on the portal wing surface, the protein sequence of portal gp91 was queried against *Herelleviridae* using PSI-BLAST, then a multiple sequence alignment of portals of *Herelleviridae* representative species was generated with MAFFT[88], and a guide tree using Phylo.io 1.0[89]. Secondary structure predictions of phage proteins were carried out with RaptorX-Property[90]. Structural predictions of portal monomers of five species spanning across *Herelleviridae* clusters were generated using AlphaFold2 and aligned to the structure of the phage 812 portal dodecamer. The electrostatic surface potential of atomic models was calculated with the UCSF ChimeraX 1.9[91] *coulombic* command, and the lipophilicity potential with the command *mlp*, using the default settings. Molecular interfaces were inspected with PDBe-PISA 1.48[82], and hydrogen bonds and salt bridges were annotated based on the intermolecular distance between donor and acceptor atom centers scoring below 4.0 Å[92]. DNA minor groove widths were measured as cross-strand $P_i$-to-$P_{i+4}$ separations reduced by $2 \times 2.9$ Å, corresponding to two phosphate van der Waals radii[93]. A circular 120 bp-long dsDNA model refined in Phenix was used for the calculation of the minor groove width of the anchor DNA bound to the portal complex, and values were averaged over 10 bp repeats. To calculate the minor groove width of the neck channel DNA, an ideal B-form dsDNA spanning the channel from the adaptor β-hairpin tube to the stopper was fitted into the density and refined in real-space in Coot. Curvature and base pair inclination of the DNA were calculated using Curves + [94]. The estimation of hydrogen bond lengths between Lys138 NZ of the portal protein and minor groove base edges of the two closest base pairs of the anchor DNA was performed by placing the refined atomic model into the C12-symmetrized map, mutating the two base pairs to all 16 Watson-Crick nucleotide combinations, and running a short minimization for each in ISOLDE at 20 °C in an Amber14 force-field, repeating the process three times. Portal crown tilt in the genome release intermediate was measured by rigidly fitting two identical crown dodecamers (residues 437–507) into the C12-symmetrized and the asymmetric map using UCSF Chimera 1.15[95]. One dodecamer was then aligned on the second dodecamer by Matchmaker, and the reported transformation matrix was used to calculate pivot point, rotation axis and XZ plane rotation angle. Figures were generated in UCSF Chimera 1.15[95], UCSF ChimeraX 1.9[91], ACD/ChemSketch 2022.2.0[96], and R 4.1.2[63].

## Statistics and reproducibility

Cryo-EM data were collected on two independent samples, with 30,553 resp. 15,371 acquired images. For resolution estimation of cryo-EM maps, particle datasets were randomly split into two subsets equal in size and reconstructed independently. DNA sequencing and subsequent analysis were performed using established workflows detailed in the Methods section. The DNA of the phage was sequenced using two independent samples and two different platforms. Using the built-in statistics in Geneious Prime v2025.0.3, the coverage of the consensus sequence of phage 812 strain K1/420 (KJ206563.2) was 607.0× (SD 111.7) for Illumina sequencing reads (NCBI SRA ID SRR36294882), and 3,013.7× (SD 2,318.2) for Oxford Nanopore sequencing reads (NCBI SRA ID SRR36291690). Residual DNA in empty capsids was analyzed using a set of 332,896 Oxford Nanopore sequencing reads (NCBI SRA ID SRR33751657).

## Reporting summary

Further information on research design is available in the Nature Portfolio Reporting Summary linked to this article.

## Data availability

The GenBank/ENA/DDBJ accession number of the phage 812 strain K1/420 genome is KJ206563.2. The whole-genome sequencing reads (NCBI BioSample ID SAMN53543416) and residual phage DNA sequencing reads (NCBI BioSample ID SAMN48784668) were deposited in the NCBI under the BioProject ID PRJNA1269267. Cryo-EM density maps are available in the Electron Microscopy Data Bank under accession IDs EMD-18048, EMD-18065, EMD-18213, EMD-18369, EMD-18372, EMD-18385, EMD-

18395, EMD-18445, EMD-18462, EMD-18489, EMD-18516, EMD-18912, and EMD-18919. Atomic coordinates and X-ray structure factors are available in the Protein Data Bank under accession IDs 8Q01, 8Q1I, 8Q7D, 8QEM, 8QEK, 8QGR, 8QJE, 8QKH, 8R5G, and 8R69. Mass spectrometry data were deposited to the ProteomeXchange Consortium *via* PRIDE[97] partner repository under dataset identifier PXD071586. Unedited image of SDS-PAGE of phage 812 is included in Supplementary Information (Fig. S27). Numeric source data for plots are supplied as Supplementary Data 1, and processed mass spectrometry data of phage 812 as Supplementary Data 2. All other data are available from the corresponding author on reasonable request.

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

## Acknowledgements

We gratefully acknowledge the Cryo-electron Microscopy and Tomography Core Facility and Proteomics Core Facility of the CEITEC MU of CIISB, Instruct-CZ Centre, supported by the Ministry of Education, Youth and Sports of the Czech Republic (MEYS CR) infrastructure project LM2023042, and the European Regional Development Fund - Projects "UP CIISB" (No. CZ.02.1.01/0.0/0.0/18_046/0015974) and "Innovation of Czech Infrastructure for Integrative Structural Biology" (No. CZ.02.01.01/00/23_015/0008175). Computational resources were provided by the e-INFRA CZ project (ID: 90254), supported by MEYS CR. This work was supported by the project National Institute of Virology and Bacteriology (Programme EXCELES, ID Project No. LX22NPO5103) - Funded by the European Union - Next Generation EU, and the project New Technologies for Translational Research in Pharmaceutical Sciences / NETPHARM, project ID OP JAC CZ.02.01.01/00/22_008/0004607, which is co-funded by the European Union. This work also received funding from ERC Consolidator Grant No. 101043452 to P.P., and from the Ministry of Health of the Czech Republic in cooperation with the Czech Health Research Council under project No. NU21J-05–00035 to T.B.

## Author contributions

M.B., M.S., Z.C. and P.B. purified the phage samples. Z.C., J.N. and M.S. performed cryo-EM data collection and processing, and Z.C. and T.F. performed cryo-EM data analysis. B.P. cloned, purified, and crystallized the stopper protein. B.P. and P.P. collected and processed crystallographic data. Z.C. carried out X-ray data analysis, X-ray and cryo-EM structural determination, deposition of atomic coordinates and maps, bioinformatic analysis and figure preparation. T.B. performed the sequencing and assembly of the phage 812 genome, and sequencing and analysis of residual capsid DNA after genome ejection. T.B. and R.P. performed the genome annotation. P.P. and R.P. designed and supervised the research. Z.C. and P.P. wrote the manuscript. All authors participated in critical reading of the manuscript and approved the final version.

## Competing interests

The authors declare no competing interests.
