## [Transparent Peer Review file · Communications Biology]

Genome anchoring, retention, and release by neck proteins of Staphylococcus phage 812

Corresponding Author: Dr Pavel Plevka

Version 0:

Reviewer comments:

Reviewer #1

(Remarks to the Author)
Review of Cienikova et al

This paper by Cienikova et al describes the structures of the neck of *S. aureus* phage phi812, a large and complex myovirus. The team previously published the high-resolution structure of the phi812 baseplate and here they are focusing on the neck region of the same virus. The structure builds on an earlier structure from the group (Novacek 2016) that also showed the neck; however the earlier study was at low resolution, precluding atomic modeling of the proteins.

In addition to various focused cryo-EM reconstructions they also present the crystal structure of the stopper protein. Comparison between the stopper alone in the crystal and the virion provides insights into how this protein prevents the DNA from coming out of the capsid after packaging. This experimental verification also confirms what has been observed for this protein in some other phages, including SPP1 and 80alpha. The presence of neck decoration proteins is an interesting detail, as is the presence of ordered DNA around the portal crown domain.

Altogether, this is extensive work, and a valuable contribution to the field. The manuscript is quite well illustrated. However, some of the structures are at relatively low resolution for model building ($>4\text{\AA}$), especially in the peripheral regions. Inside the tail the resolution is 6\AA at best. A local resolution map is only available for the empty capsid (Fig. S15). Similar maps should be shown for the other reconstructions as well.

This is especially true for the inside of the tail. The density shown in Fig 7 is clearly not at sufficient resolution to conclusively say that the structure below the DNA is the TMP. Could this density instead be a tail completion protein (TCP), which was shown in 80alpha to sit inside the tail between the DNA and the TMP? Does phi182 have a recognizable TCP gene? If so, what is its (predicted) structure? If it does not have a TCP, why not? This should be discussed in the manuscript.

How was the "genome release intermediate" produced? Why is DNA ejection halted? How does this compare to an actual infection when presumably the whole DNA should get ejected?

The paper is concisely written, which is a good thing; however, the discussion is rather brief—little more than a summary of the results—and the authors could do a better job of discussing their results in the context of the many other phages for which neck structures have been determined and other models for DNA packaging and ejection. What are the similarities and differences? Do their structures support or contradict previous results? How similar is phi812 to phage K?

Figure 1 shows a space filling model of the atomic model. Here or elsewhere they should show the actual electron density map. This could be a composite if that works best, and could be segmented and colored by protein, but it should show the actual experimental density.

Line 62-53: The central channel... is a bit awkwardly phrased. Rewrite to something like "In the absence of attachment to the tail terminator, the central channel of the stopper protein hexamer is closed by gating loops. In the virion, where the stopper is attached to the tail terminator, the central channel is open..."

Line 331: "It has been proposed..." This is well established at least for lambda, so maybe a stronger statement like "The

TMP acts as a scaffold for tail tube assembly in phage lambda." Whether it is a trimer, rather than a monomer or hexamer, in all cases, however, is still a point of contention.

The Kizziah 2025 reference is lacking the full citation: Structure 33, 1063-1073.e2.

Reviewer #2

(Remarks to the Author)

The manuscript (MS) by Z. Cienkova and co-authors entitled "Genome anchoring, retention, and release by neck proteins of Staphylococcus phage 812" describes nicely structural features of the neck apparatus of the bacteriophage in the genome containing and genome-release intermediate states. Providing a detailed structural account, the MS is overall well-written, with novel findings and provides key information related to the comparison between different phages with contractile tails. In the context of biological mechanism by which tailed phages assemble/release the packaged genome, the authors propose a mechanism by which the neck apparatus contributes to genome ejection by phages. The authors have found some interesting structural features, that were not described in the previous publications. As it has been expected, new studies based on advanced methods of the modern structural cryogenic electron microscopy provide a lot of new information, revealing novel structural details of macromolecular complexes, and at the same time, raise new questions that possibly could be addressed by following specific structural and biochemical means to make the structural account stronger related to our understanding of the biological mechanism.

While the manuscript has addressed many points linked to the structure/function relationship, some of them need additional explanations and at times, additional clarifications. The number of questions listed below are related to the structural analysis described in MS which should be addressed.

The authors can find the questions raised by this interesting study below:

1. Role of the decoration proteins: The authors did not discuss in a transparent way what was the role of the decoration's proteins. It was found that they are attached to the phage (neck area) in the fully assembled phage but absent in the phage that is in the intermediate state, where a part of genome has been ejected. It remains unclear if the phage will be self-assembled completely under conditions where the decoration proteins will be removed in the mutated sequence. Is it possible that they play a role of chaperones that are essential for the successful self-assembly of phages but not for their activity?
2. What is the biological significance of channel dimensions of the adaptor observed (narrower at top and wider at the base connecting to stopper)? Is the adaptor in an open/closed/intermediate state:
3. Lines 145-146: It is difficult to follow why the authors solved a purified stopper protein structure when the neck assemblage complex described prior had the stopper protein described? It is difficult to follow whether the conformation of the helices observed is a result of crystal lattice packing or in fact due to the lack of tail/DNA. What is the sequence/structural similarity of phage 812 stopper protein to phage SPP1 stopper protein?
4. Lines 154-170: The discussion around open and closed conformation of stopper protein based on crystal structure and neck complex stopper protein structure seems difficult to follow as it is not a direct comparison between two neck complexes captured in open and closed states. Did the authors analyse the neck complex structures from a) genome filled, b) genome-released partly state, and c) empty states of the phage to make comparisons, that would make a compelling argument for structural changes proposed as a mechanism.
5. Interesting to note that the in vitro induced genome released state has stopper protein N & C termini disordered. Why could it be and how this information has been obtained?
6. Lines 171-189. It is really confusing: why the authors discussed the idea that the DNA release can be triggered by changes in the neck complex of the phage? It is not consistent with the accepted idea: the release is initiated by the signal that is produced by the tail tip recognition of the specific receptor incorporated into the outer cell surface of a host cell. As soon the host cell has been recognised the tail became attached to the cell and signal is transferred along the tail measure protein towards the phage capsid and some changes take place within the neck, the channel becoming open and under the inner pressure within the capsid the genome is released into the host cell. The authors have to justify their idea that the conformational changes in the neck trigger the DNA release: when and how?
7. Lines 191-195. There are some mix-ups related to the interactions of the tail terminator and the tail major protein upper ring that is in the contact with the neck complex. The authors write at the beginning of this paragraph, that the terminator interacts with the tail sheath proteins, however after contraction the terminator does not change its conformation and apparently it interacts only with the major tail protein and not with the sheath proteins. Please clarify this paragraph.
8. Lines 234-238. How is the tilt angle of the crown funnel been measured? Since this is an interesting fact, some more details are required. It would be good to define what do the authors name as an "anchor" DNA ring. What does it anchor? How reliable was this measurement at a resolution of the asymmetrical reconstruction? Did it depend on the (presumably symmetrised) starting reconstruction or was the map calculated ab-initio? Also, what was a distance between wing domains of the adjacent monomers in the portal protein complex resolved? How does it correspond to the one turn of the dsDNA (lines 209-210)? Of which turn (A or B form of the dsDNA) are the authors are talking?
9. Lines 297-301. There is some confusion here: how can the authors explain the effect of the pausing of the DNA ejection? Is it related to the canonical B to A-form tilt at the segment the authors describe? What is the biological role and what

induces the stalled to the non-stalled state? What sort of evidence from the data justifies this statement? Could the authors provide some hypothesis why and how the composition of the translocating DNA can cause the effect of stalling DNA within the channel, and which factors play a role in continuation of the DNA ejection?

10. Lines 296-301 and 313-329. The authors have analysed two states of the phage: fully filled with the DNA and in partly filled capsid (intermediate state?). Have the authors made a comparison of the DNA states within adaptor in these two forms of the phage? The pressure of the DNA on the neck complex was less in the second case. Did the authors observe any changes in the DNA conformation that is became mostly of the B-form? Are there any other evidence that could support this hypothesis? Can the authors provide details such as base-pair inclination and bend angle at the 'kink' where the A-form configuration is suggested? It seems that the interface at adaptor which induces the DNA kink has promoter-like properties to interact and induce DNA kink. Did the authors consider mutation at the interface to switch properties of DNA bending? More close-up details (figure panels) to show interface between bent DNA and adaptor protein inducing that feature would be helpful.

11. Lines 346-362. Conclusions: it would be recommended that conclusions should be modified according to the changes of the text after answering on general comments. They have to be consistent: a) it was not clear how the portal protein "influences DNA spooling inside the head"; b) A structure of the stopper proteins within fully packaged capsid was not shown: DNA goes through the entire neck channel up to the upper part of the tail, so it does not block the neck channel exit with their gating loops, the structure with the blocked channel was not provided.

Minor comments:

Lines 60-62? What was the strategy for selecting phages from cryo-micrographs for processing of neck intermediate structures? Did the authors also study empty phage neck structures (where the genome is fully released)?

Lines 71-72: Clarification is needed on which 'state' of phage is this structure derived from. Presumably the genome filled state?

Lines 81-84: Did the authors resolve the entire portal complex as usually the crown domain is less well resolved? One note would be to show a monomer structural model with different parts labelled (with amino acids of N and C termini) to show resolvability and structural conservation (with related phage portal proteins) of the different domains?

Line 177: Figure 1D is not obvious that is from the genome release intermediate neck apparatus. Also fig. S14A-B do not show the described disordered regions of stopper protein = a cut-away view in S14B would be useful. A figure to show this shift would be helpful.

Lines 180-181: The RMSD value seems smaller than the stated resolution. Are these changes significant?

Lines 186-187: The authors describe a comparison of empty vs. filled phage neck protein complexes showing no difference; This has not been described earlier or shown in the figure. What about the comparison of empty vs. genome-release intermediate? Given also the authors describe a DNA kink in the genome release intermediate at adaptor protein and a second DNA genome observed on the wing domain of portal complex in both genome-filled and genome release intermediate, some structural changes would be expected when the DNA is no longer there in both cases?

Line 204: Fig.S17 – the authors show the asymmetric map (in teal) and 12-fold symmetrised map in grey. How is the double-stranded DNA density well resolved (with major and minor grooves clearly well resolved?) in the grey map if the 3D map calculated was symmetrised by applying C12 symmetry? Are the authors implying the DNA is also 12-fold symmetrically arranged? Kindly elaborate.

Lines 205-207: Isn't this encircled dsDNA segment also present in the genome-filled (virion) neck complex – albeit not well resolved in the symmetrised structure (Figure 1), if so, what are the differences between 'states' of this dsDNA.

Lines 209-211. Please explain the following sentence (it sounds somewhat confusing): "...the DNA double helix makes one turn per portal monomer...". What is the conformation of this circular dsDNA around the portal complex? Given the state that has been captured of the portal complex with encircled DNA at wing and crown domain. Can the authors comment on any local conformational changes at the portal which may be present (when compared with portal proteins of similar phylogeny) to accommodate the encircled dsDNA?

Line 218-219: What do the authors mean by 'inter-particle' averaging? Did the authors try masked classification to see if they can separate different subsets of particles to improve DNA density? Based on the DNA mapping of the segment bound to portal – can the authors comment on whether this is sub-fragment of DNA that is independent to the genome packaged inside the capsid? How are the two connect as based on the structures they seem independent (both in virion and genome-release intermediate?).

Lines 254-255. English has to be corrected: "...loop interacts with the crown base of the same portal protein (explain what is meant by "the same portal protein?"). Are the authors discussing interactions within one subunit?

Line 284-296 – The bent DNA at portal-adaptor junction also present in the virion map (Fig. S19 B); do the authors imply that

in two states of the phage, the DNA is stalled in a similar way whereby the identity of the DNA segment present in virion and genome-release is the same? What would be the biological reason for having a stalled DNA state in the virion (as DNA is fully packaged in the capsid)?

Line 317. What is it “the right end of phage 812”., The term “right” is unclear , it is confusing this given context.

Version 1:

Reviewer comments:

Reviewer #1

(Remarks to the Author)

Thank you for your careful consideration of my comments and suggestions. I find that the revisions adequately address my concerns.

Reviewer #2

(Remarks to the Author)

The manuscript by Z. Cienikova and co-authors entitled “Genome anchoring, retention, and release by neck proteins of Staphylococcus phage 812” address structural features of the neck apparatus of the bacteriophage in the genome containing and genome release intermediate states, ascribed to genome retention and release. The authors have carefully addressed the major and minor points addressed with satisfactory conclusions as well as highlighting limitations of the study. With the few suggested comments below, the manuscript is suitable to be published.

Minor points:

Line 187-190: ‘The crown of the portal complex moves as a rigid body (RMSD of 1.4 Å over 804 C α atom pairs after crown alignment).’ Based on the cryo-EM derived model (from 4 Angstrom map), reporting a ‘change’ of 1.4 Å does not seem comparable on the level of cryoEM derived model. The authors can simply say ‘minimal change was observed when crown domain models are compared.’ The authors also later mention the crown funnel tilts by 3.5 degrees (Line 261-262), based on interactions with DNA loop. I think some clarification would help make this point strong that some local changes are induced but no large changes were observed as a result of DNA present in the channel.

Given no major changes were observed between genome containing and genome-release intermediate, it can be attributed to the ‘stalled’ state of the complex?

For all graphs depicting a single point value (e.g., mean) with error bars, you must add individual data points or convert the graph to a boxplot or dot-plot to show data distribution.

A: Datapoints were added to Fig. 4F.

It's mandatory to provide access to the numerical source data for graphs and charts either through a repository or by providing the data in a Supplementary Data file (in excel format).

A: Source data for Figs. 4F, 6E and S12 are included as tables in Supplementary_data.xlsx. Data for Figs. 4G, S18C and S24 were deposited as NCBI BioProject. Data for Figs. S3 and S15 were deposited in PDB.

All blots/gels must be accompanied by size markers in every figure panel. Uncropped and unedited blot/gel images must be included as Supplementary Figure(s) in the Supplementary Information pdf.

A: Gel image from Fig. S26B is now shown unedited in new Fig. S27.

Please ensure that you have complied with the data deposition policies at the Nature Portfolio, please see here.

A: Cryo-EM and X-ray data were deposited in EMDB and PDB, and DNA sequencing data in GenBank and NCBI BioProject DB. This is stated in the “Data availability” section of the manuscript (lines 614-621).

Please ensure that you have complied with our policies on research involving animals and humans, see here.

A: This work does not include research involving animals or humans.

Please follow the ARRIVE guidelines for reporting animal experiments. Please fully complete an ARRIVE checklist including both the essential and recommended set of items (adding information to the manuscript where needed) and upload this with your revised manuscript.

A: This work does not include research involving animals.

Please also see our revision checklist for guidance on formatting the manuscript and complying with our policies. A comprehensive guide to our formatting requirements for final submissions is also available for your reference here.

A: Revision checklist has been completed and is included.

Note from Authors: Unless otherwise stated, line, figure and table numbers refer to the updated version of the manuscript with tracked changes.

Reviewers' comments:

Reviewer #1 (Remarks to the Author):

Review of Cienikova et al

This paper by Cienikova et al describes the structures of the neck of *S. aureus* phage phi812, a large and complex myovirus. The team previously published the high-resolution structure of the phi812 baseplate and here they are focusing on the neck region of the same virus. The structure builds on an earlier structure from the group (Novacek 2016) that also showed the neck; however the earlier study was at low resolution, precluding atomic modeling of the proteins.

In addition to various focused cryo-EM reconstructions they also present the crystal structure of the stopper protein. Comparison between the stopper alone in the crystal and the virion provides insights into how this protein prevents the DNA from coming out of the capsid after packaging. This experimental verification also confirms what has been observed for this protein in some other phages, including SPP1 and 80alpha. The presence of neck decoration proteins is an interesting detail, as is the presence of ordered DNA around the portal crown domain.

1.1. Altogether, this is extensive work, and a valuable contribution to the field. The manuscript is quite well illustrated. However, some of the structures are at relatively low resolution for model building ($>4\text{\AA}$), especially in the peripheral regions. Inside the tail the resolution is 6\AA at best. A local resolution map is only available for the empty capsid (Fig. S15). Similar maps should be shown for the other reconstructions as well.

A: Local resolution maps for all deposited reconstructions have now been included as Fig. S5 for the virion, and Fig. S17 for the genome release intermediate, replacing original Fig. S15.

This is especially true for the inside of the tail. The density shown in Fig 7 is clearly not at sufficient resolution to conclusively say that the structure below the DNA is the TMP. Could this density instead be a tail completion protein (TCP), which was shown in 80alpha to sit inside the tail between the DNA and the TMP? Does phi182 have a recognizable TCP gene? If so, what is its (predicted) structure? If it does not have a TCP, why not? This should be discussed in the manuscript.

A: No homologue of phage 80alpha TCP could be identified in phage 812, whether by sequence-based (PSI-BLAST) or structure-based (Foldseek, Dali) approaches. The two phages are from different families. Our work shows that phage 812 genomic terminus extends into the tail channel (Fig. 7) and unlike 80alpha, phage 812 does not contain any protein inside the neck channel formed by HTCP and HTJP in 80alpha, equivalent to adaptor and stopper proteins in phage 812.

We additionally catalogued all genes with unknown function belonging to the neck or tail gene clusters of phage 812 genome. We checked their presence in MS of purified virions and fitted their predicted tertiary structures into phage 812 tail channel density reconstructed in C1. The most promising candidate (gp106, 103 residues-long, gene located between tail tube and predicted tail assembly chaperone genes, gp present in MS at 14% coverage) fits the density with a cross-correlation of 0.6, compared to CC 0.8 for the dsDNA model and 0.9 for the trimeric TMP model shown in Fig. 7. All taken together, we conclude that TMP is the most likely candidate for the density bordering dsDNA inside the tail channel, as stated previously (lines 344-346). However, we agree that the achieved map resolution does not warrant a discussion about the structure of TMP N-terminus and we removed it from the manuscript (lines 347-350) and from Fig. 7 (panel D).

We included the comparison to phage 80alpha in the manuscript as follows (lines 350-352): “Unlike phage 80 α (Kizziah, 2025), phage 812 does not contain a tail completion protein blocking DNA inside the neck channel, pointing at different genome gating mechanisms used by these two phages.”

1.2 How was the “genome release intermediate” produced?

A: We have now included the description in the section “Genome release is not dependent on changes in neck structure” (lines 179-182): “We induced genome release in purified phage 812 virions by in vitro treatment with urea. The capsids showed a continuum of residual genome density, as the mechanisms that ensure completion of DNA ejection in vivo cannot occur. We selected capsids with the lowest average inner density, and we termed this dataset the genome release intermediates.” We also added more details about particle selection to the flowchart in Fig. S14 (upper right inset).

Why is DNA ejection halted?

A: The DNA ejection occurred *in vitro*; thus, the mechanisms that ensure completion of DNA ejection *in vivo* cannot occur. We have now included this explanation in the manuscript (lines 179-181): “The capsids showed a continuum of residual genome density, as the mechanisms that ensure completion of DNA ejection *in vivo* cannot occur”.

How does this compare to an actual infection when presumably the whole DNA should get ejected?

A: There are various hypotheses about the mechanisms ensuring completion of phage genome delivery: flow of water and/or ions into the phage particle and then through the tail channel into the bacterial cell, activity of RNA polymerases or phage ejectome proteins, or DNA diffusion in the bacterial cytoplasm. We added the explanation to the manuscript (lines 338-339): “Pull on the stalled DNA caused by its diffusion in the bacterial cytoplasm or by the activity of RNA polymerase could resume the genome ejection (Molineux & Panja, 2013; Chen, 2018)”.

1.3 The paper is concisely written, which is a good thing; however, the discussion is rather brief—little more than a summary of the results—and the authors could do a better job of discussing their results in the context of the many other phages for which neck structures have been determined and other models for DNA packaging and ejection. What are the

similarities and differences? Do their structures support or contradict previous results? How similar is phi812 to phage K?

A: Phage 812 belongs to the same genus of Kayviruses as phage K, and is the only structurally characterized phage in its family. As the reviewer notes, there are a number of distantly related phages with solved neck structures but only a few of them in different stages of life cycle, allowing to hypothesize about DNA packaging and retention. We discussed the role of portal proteins in anchoring DNA based on portal wing charges (lines 87-93 and Figs. S8-9), compared phage 812 and SPP1 stopper proteins and their function in genome retention in tailless capsids (lines 156-160, Fig. S13), and compared phage 812 with phages ϕ 29 and lambda regarding the genome arrangement in the adaptor and the tail tube (lines 326-329, 364-366). We have now included a discussion on differences in genome gating between 812 and 80alpha virions (lines 350-352: “Unlike phage 80 α (Kizziah, 2025), phage 812 does not contain a tail completion protein blocking DNA inside the neck channel, pointing at different genome gating mechanisms used by these two phages.”), and a comparison to phage E217 about the influence of tail sheath contraction on the neck structure (lines 214-217, 226-229): “Inter-disc β -sheet complementation was observed also in phages E217, Pam3 and Pa193 and appears to be a conserved feature of phages with contractile tails (Li, 2023; Yang, 2023; Iglesias, 2024). The tail terminator protein of phage 812 is unique in having an outer β -sandwich domain that clamps tail sheath domain I against the tail tube (Fig. 1). [...] In comparison, following the tail contraction of phage E217, domains I of neck-adjacent tail sheath proteins move away from the gateway protein gp29, a homologue of phage 812 tail terminator, as gp29 lacks a β -sandwich domain locking the tail sheath domain I. The C-terminal β -strand of gp29 swings outwards to accommodate the conformational change (Li, 2023)”.

1.4 Figure 1 shows a space filling model of the atomic model. Here or elsewhere they should show the actual electron density map. This could be a composite if that works best, and could be segmented and colored by protein, but it should show the actual experimental density.

A: Fig. 1B shows the cryo-EM density of the virion neck and Fig. S18 of the genome release intermediate. We have now included figures of model-colored composite maps of the neck of the virion and the genome release intermediate as Figs. S4 and S16, with the following caption: “Fig. S4/S16. Composite map of the neck of phage 812 virion/genome release intermediate. The unmasked map is shown as gray transparent surface. Map zones within a radius of 3 Å of fitted atomic models are colored according to the protein: anchor DNA in hues of red, portal in light blue, adaptor in gold, stopper in magenta, tail terminator in orange, stopper decoration in hues of forest green, terminator decoration in hues of steel blue, tail tube in hues of violet, and tail sheath in hues of desaturated green.”.

1.5 Line 62-53: The central channel... is a bit awkwardly phrased. Rewrite to something like “In the absence of attachment to the tail terminator, the central channel of the stopper protein hexamer is closed by gating loops. In the virion, where the stopper is attached to the tail terminator, the central channel is open...”

A: We accept the suggestion, thank you (lines 63-67).

Line 331: “It has been proposed...” This is well established at least for lambda, so maybe a stronger statement like “The TMP acts as a scaffold for tail tube assembly in phage lambda.” Whether it is a trimer, rather than a monomer or hexamer, in all cases, however, is still a point of contention.

A: We accept the suggestions as follows (lines 370-371): “The TMP complex acts as a scaffold for tail tube assembly”.

The Kizziah 2025 reference is lacking the full citation: *Structure* 33, 1063-1073.e2.

A: Thank you, this has now been corrected (line 874): “Kizziah, J. L., Mukherjee, A., Parker, L. K., & Dokland, T. (2025). Structure of the *Staphylococcus aureus* bacteriophage 80α neck shows details of the DNA, tail completion protein, and tape measure protein. *Structure*, 33(6), 1063-1073.e2. <https://doi.org/10.1016/j.str.2025.03.007>”

Reviewer #2 (Remarks to the Author):

The manuscript (MS) by Z. Cienikova and co-authors entitled “Genome anchoring, retention, and release by neck proteins of *Staphylococcus* phage 812” describes nicely structural features of the neck apparatus of the bacteriophage in the genome containing and genome-release intermediate states. Providing a detailed structural account, the MS is overall well-written, with novel findings and provides key information related to the comparison between different phages with contractile tails. In the context of biological mechanism by which tailed phages assemble/release the packaged genome, the authors propose a mechanism by which the neck apparatus contributes to genome ejection by phages. The authors have found some interesting structural features, that were not described in the previous publications. As it has

been expected, new studies based on advanced methods of the modern structural cryogenic electron microscopy provide a lot of new information, revealing novel structural details of macromolecular complexes, and at the same time, raise new questions that possibly could be addressed by following specific structural and biochemical means to make the structural account stronger related to our understanding of the biological mechanism. While the manuscript has addressed many points linked to the structure/function relationship, some of them need additional explanations and at times, additional clarifications. The number of questions listed below are related to the structural analysis described in MS which should be addressed.

The authors can find the questions raised by this interesting study below:
2.1. Role of the decoration proteins: The authors did not discuss in a transparent way what was the role of the decoration's proteins. It was found that they are attached to the phage (neck area) in the fully assembled phage but absent in the phage that is in the intermediate state, where a part of genome has been ejected. It remains unclear if the phage will be self-assembled completely under conditions where the decoration proteins will be removed in the mutated sequence. Is it possible that they play a role of chaperones that are essential for the successful self-assembly of phages but not for their activity?

A: We have been developing approaches to genetically modify phage 812 using CRISPR/Cas and phage boot-up tools, but without success. Therefore, we are not able to engineer mutant phages that do not express the neck decoration proteins. However, we agree with reviewer #2's suggestion that the proteins may function as neck assembly chaperones. We have now included this speculation in the manuscript (lines 146-148): "It is also possible that the neck whisker and decoration proteins play a role of chaperones that are important for a successful self-assembly of phage 812 virions."

2.2. What is the biological significance of channel dimensions of the adaptor observed (narrower at top and wider at the base connecting to stopper)? Is the adaptor in an open/closed/intermediate state:

A: We did not observe any differences in the width of the adaptor complex channel between the 812 virion and the genome release intermediate (Fig. 1D). Therefore, there is no evidence that the complex exists in more than one state. The geometry of the adaptor complex enables neck assembly and genome translocation. The wide top enables the insertion of the base of the portal complex formed by the clip domains (Fig. 1). The narrow base of the adaptor is, in turn, inserted into the stopper protein channel. The diameter of the adaptor base (30 Å) is conserved among long-tailed phages, and this width is probably optimized for the translocation of straight B-form dsDNA (we now included this commentary into the manuscript, lines 99-102, as follows: "The diameter of the adaptor β -hairpin tube channel is remarkably conserved among both siphon- and myophages with known structures (Orlov, 2022; Ayala, 2023; Li, 2023; Yang, 2023; Gu, 2024; Iglesias, 2024; Li, 2025; Kizziah, 2025) and is probably optimized for the translocation of straight B-form dsDNA").

2.3. Lines 145-146: It is difficult to follow why the authors solved a purified stopper protein structure when the neck assemblage complex described prior had the stopper protein described? It is difficult to follow whether the conformation of the helices observed is a result

of crystal lattice packing or in fact due to the lack of tail/DNA. What is the sequence/structural similarity of phage 812 stopper protein to phage SPP1 stopper protein?

A: The structure of a tailless neck assemblage complex of phage 812 is not available. Therefore, the structure of the stopper protein complex not embedded within the neck is of interest because of its putative function of holding the genome inside a particle before the tail attachment. The stopper protein is a hexamer in solution and in the crystal (Fig. S12). The gating α -helices are located inside the hexamer channel and do not participate in inter-hexamer crystal contacts. Phage 812 and SPP1 stopper proteins have 9% sequence identity and a C α -RMSD of 6.1 Å. The structural comparison is shown in Fig. S13A-B, and we now included the sequence identity and RMSD scores in the caption as follows: “The stopper proteins of phages SPP1 and 812 have 9% sequence identity, and an RMSD of 6.1 Å over 42 matched C α atoms belonging to the core β -barrel and the gating loop”.

2.4. Lines 154-170: The discussion around open and closed conformation of stopper protein based on crystal structure and neck complex stopper protein structure seems difficult to follow as it is not a direct comparison between two neck complexes captured in open and closed states. Did the authors analyse the neck complex structures from a) genome filled, b) genome-released partly state, and c) empty states of the phage to make comparisons, that would make a compelling argument for structural changes proposed as a mechanism.

A: The structures of stopper proteins of phage 812 are available for the virion (genome-filled state) and the genome release intermediate (state with partly released genome). They are near-identical and in the open conformation. This indicates that the stopper protein structure does not change upon genome ejection (as described in the section titled “Genome release is not dependent on changes in neck structure”, line 178). In contrast, the crystal structure of the isolated stopper protein hexamer has a closed channel. This is stated in the manuscript on lines 155-160.

2.5. Interesting to note that the in vitro induced genome released state has stopper protein N & C termini disordered. Why could it be and how this information has been obtained?

A: The densities of these termini are not resolved in the cryo-EM map of the genome release intermediate, indicating their mobility. We speculate that the loss of the structure is due to the detachment of neck decoration proteins that bind the termini in the virion (lines 184-188).

2.6. Lines 171-189. It is really confusing: why the authors discussed the idea that the DNA release can be triggered by changes in the neck complex of the phage? It is not consistent with the accepted idea: the release is initiated by the signal that is produced by the tail tip recognition of the specific receptor incorporated into the outer cell surface of a host cell. As soon the host cell has been recognised the tail became attached to the cell and signal is transferred along the tail measure protein towards the phage capsid and some changes take place within the neck, the channel becoming open and under the inner pressure within the capsid the genome is released into the host cell. The authors have to justify their idea that the conformational changes in the neck trigger the DNA release: when and how?

A: We agree that phage 812 genome release is not triggered by conformational changes in the neck structure, as stated in the section titled “Genome release is not dependent on changes in neck structure” (line 178). As the word “trigger” used originally in the section title may have been confusing (implying the first event in a causal chain), we replaced it by

the term “depend on”. The neck structures of the 812 virion and the genome release intermediate are nearly identical (Fig. 1D). Therefore, it is likely that the genome release is enabled by the release of tail tape measure proteins. This is described in detail in our manuscript section “Genomic end and the tail tape measure protein” (lines 340-347).

2.7. Lines 191-195. There are some mix-ups related to the interactions of the tail terminator and the tail major protein upper ring that is in the contact with the neck complex. The authors write at the beginning of this paragraph, that the terminator interacts with the tail sheath proteins, however after contraction the terminator does not change its conformation and apparently it interacts only with the major tail protein and not with the sheath proteins. Please clarify this paragraph.

A: The tail of phage 812 is formed by tail sheath and tail tube proteins. The term major tail protein is not used for phages with contractile tails. Nevertheless, to make the text easier to follow, we have now expanded the description of the interactions of tail terminator proteins with tail tube proteins (lines 203-209). The interactions of tail terminator proteins with tail sheath proteins are described extensively (lines 210-226).

“The tail terminator complex caps the tail and interacts with the terminal hexamers of tail tube and tail sheath proteins. The core β -sheets of six tail terminator proteins build a β -barrel of the same size and shape as the tail tube β -barrel, extending the tail channel seamlessly into the neck (Fig. S6 D and H). The interaction is strengthened by N-termini of tail tube proteins which insert into gaps between cores and β -sandwich domains of tail terminator proteins. Tail sheath contraction and genome ejection have no effect on the conformation of the terminal tail tube hexamer nor on its interface with the tail terminator complex (Fig. 1D).”

2.8. Lines 234-238. How is the tilt angle of the crown funnel been measured? Since this is an interesting fact, some more details are required.

A: We have now included the description of the calculation of the crown funnel tilt angle in the Materials and Methods section, lines 596-600: “Portal crown tilt in the genome release intermediate was measured by rigidly fitting two identical crown dodecamers (res. 437-495) into the C12-symmetrized and the asymmetric map using UCSF Chimera 1.15 (Pettersen, 2004). One dodecamer was then aligned on the second dodecamer by Matchmaker, and the reported transformation matrix was used to calculate pivot point, rotation axis and XZ plane rotation angle.”

It would be good to define what do the authors name as an “anchor” DNA ring. What does it anchor?

A: Thank you, we have now included a definition of the anchor DNA ring in the manuscript, lines 231-232: “A section of dsDNA, which we named ‘anchor DNA’, encircles the portal complex at the level of wing domains in the cryo-EM reconstruction of the phage 812 genome release intermediate.” We selected the name “anchor DNA” because it attaches the genome to the portal. The term has been used before (Liu, 2019).

How reliable was this measurement at a resolution of the asymmetrical reconstruction?

A: The dimensions of the crown funnel are $65 \times 75 \text{ \AA}$. Therefore, even small differences in orientation can be accurately determined at the local $4.0 - 5.5 \text{ \AA}$ resolutions of the compared maps.

Did it depend on the (presumably symmetrised) starting reconstruction or was the map calculated ab-initio?

A: The asymmetric reconstruction originated from the C12 reconstruction, as shown in Fig. S14. Unlike the rest of the portal complex, the crown funnel was poorly resolved in the twelvefold symmetrized reconstruction, indicating that it deviates from the symmetry. We added a new Fig. S17A showing the local resolution of the map.

Also, what was a distance between wing domains of the adjacent monomers in the portal protein complex resolved? How does it correspond to the one turn of the dsDNA (lines 209-210)? Of which turn (A or B form of the dsDNA) are the authors are talking?

A: The arc distance between neighboring wing domains, described by the anchor DNA at a radius of 65 Å, is 34 Å, which corresponds to one B-DNA helix turn (Sinden, 1998). This is stated on lines 233-234. We have now displayed this information in Fig. 4B, and updated its caption: “The arc distance between neighboring wing domains described by the anchor DNA is 34 Å (indicated in navy blue), which corresponds to one B-DNA helix turn.”.

2.9. Lines 297-301. There is some confusion here: how can the authors explain the effect of the pausing of the DNA ejection?

A: Genome ejection pausing has been experimentally demonstrated for other phages, and the sequencing of residual DNA inside capsids induced to release their genome shows that the pausing occurs for phage 812 *in vitro* (Fig. S24). The pausing can help to regulate phage genome expression. For example the early genes, located in the DNA region that is ejected first, can inhibit the action of anti-phage systems or degrade host genome to protect the phage DNA from degradation. We added this explanation into the manuscript text (lines 334-338): "The pausing may contribute to the regulation of phage gene expression in the initial stages of infection, help to avoid targeting by bacterial immune systems before phage counter-proteins are synthesized, and prevent phage DNA degradation during phage-induced cleavage of the host genome, as observed in phage T5 (Lanni, 1968; de Frutos, 2005)."

Is it related to the canonical B to A-form tilt at the segment the authors describe?

A: Yes, we propose that the ejection pausing is due to the DNA tilt and B-to-A conversion in the adaptor channel, as stated in the manuscript (lines 330-332): "During genome delivery, the A-form conversion in the adaptor chamber of phage 812 could provide a means to strategically pause DNA ejection [...]"

What is the biological role and what induces the stalled to the non-stalled state?

A: The pausing can help to regulate phage genome expression (Lanni, 1968). The continuation of the genome ejection could be induced by thermal motions of the DNA (both *in vitro* and *in vivo*), by pulling of the DNA into the cell by RNA polymerase or phage ejectome proteins, or by DNA diffusion in the bacterial cytoplasm (*in vivo*) (Molineux & Panja, 2013; Chen, 2018). We added this explanation into the manuscript text (lines 334-339): "The pausing may contribute to the regulation of phage gene expression in the initial stages of infection, help to avoid targeting by bacterial immune systems before phage counter-proteins are synthesized, and prevent phage DNA degradation during phage-

induced cleavage of the host genome, as observed in phage T5 (Lanni, 1968; de Frutos, 2005). Pull on the stalled DNA caused by its diffusion in the bacterial cytoplasm or by the activity of RNA polymerase could resume the genome ejection (Molineux & Panja, 2013; Chen, 2018).”).

What sort of evidence from the data justifies this statement?

A: Genome ejection pausing has been experimentally demonstrated for phage T5 (Lanni, 1968; de Frutos, 2005). In phage 812, the evidence of ejection stalling is provided by the existence of a widened A-form DNA in the adaptor chamber of particles at different stages of genome release (lines 301-307 and Fig. 6), and by the discrete lengths of residual DNA strands in the capsids of these particles (Fig. S24). The evidence for the reverse A-to-B form conversion is the ability to induce genome release in virions where the widened A-form is present (lines 312-314 and Fig. S23).

Could the authors provide some hypothesis why and how the composition of the translocating DNA can cause the effect of stalling DNA within the channel, and which factors play a role in continuation of the DNA ejection?

A: Specific DNA sequences can favor bending or B-to-A conversion in phage 812 adaptor channel. We added the following explanation to the manuscript text (lines 319-324): “AT-rich dsDNA is prone to bending under the influence of positive ions (Mirzabekov & Rich, 1979; Shui, 1998; Hancock, 2011), and GC-rich dsDNA enables B-to-A form conversion (Minchenkova, 1986; Kulkarni & Mukherjee, 2013). Therefore, an AT-rich DNA tract located in the adaptor β -hairpin tube followed by a GC-rich tract located in the adaptor chamber is susceptible to forming the structure observed in phage 812 adaptor channel.”. Continuation of the ejection may be induced by thermal motions or RNA polymerase pulling (comment added on lines 338-339: “Pull on the stalled DNA caused by its diffusion in the bacterial cytoplasm or by the activity of RNA polymerase could resume the genome ejection (Molineux & Panja, 2013; Chen, 2018).”).

2.10. Lines 296-301 and 313-329. The authors have analysed two states of the phage: fully filled with the DNA and in partly filled capsid (intermediate state?). Have the authors made a comparison of the DNA states within adaptor in these two forms of the phage?

A: Yes, we compared the two structures (lines 312-313 and Fig. S23). We concluded that the conformation of DNA is identical in both cases.

The pressure of the DNA on the neck complex was less in the second case. Did the authors observe any changes in the DNA conformation that is became mostly of the B-form? Are there any other evidence that could support this hypothesis?

A: Indeed, the DNA below the adaptor of the genome release intermediate is not under pressure and adopts the B-form (stated on line 302 and shown in Fig. 6). In contrast, the DNA in the tail tube above the putative tail tape measure proteins is in a B/A hybrid form (lines 354-364 and Fig. 7). As far as we know, there is no independent experimental evidence characterizing the state of the DNA. However, the B-form is the normal state of dsDNA, and the A-form is formed only under specific conditions such as pressure, remodeling by proteins, or dehydration (Minchenkova, 1986; Kulkarni & Mukherjee, 2013).

Can the authors provide details such as base-pair inclination and bend angle at the 'kink' where the A-form configuration is suggested?

A: There are several bends in the channel DNA in the genome release intermediate, but the most prominent are those at the A/B-form junctions and the one inside the adaptor β -hairpin tube (manifesting as a local squeezing of the minor groove, Fig. 6E). We have now included a new Fig. S22 showing the local curvature of the helical axis of the DNA and the inclination of the base pairs, with the following caption: "Geometry of the channel DNA in the genome release intermediate. Portal and adaptor complexes and the DNA are shown as cartoons. (A) DNA base pairs are colored according to local axial curvature. Prominent bending sites are located near the two B/A-form junctions and inside adaptor β -hairpin tube. (B) DNA base pairs are colored according to base pair inclination. Expected inclination angles are ca -2° for B-form DNA and 20° for A-form DNA."

It seems that the interface at adaptor which induces the DNA kink has promoter-like properties to interact and induce DNA kink. Did the authors consider mutation at the interface to switch properties of DNA bending?

A: We have been developing approaches to genetically modify phage 812 using CRISPR/Cas and phage boot-up tools, but without success. Therefore we cannot prepare mutants of phage 812.

More close-up details (figure panels) to show interface between bent DNA and adaptor protein inducing that feature would be helpful.

A: The interactions at the site of bending inside adaptor β -hairpin tube are shown in Fig. 6A viewed along the Y axis, in Fig. 6D along the Z axis, and in Fig. 6F along an axis tilted by -24 dg. The limited resolution of the reconstruction does not allow characterization of molecular details of the interaction.

2.11. Lines 346-362. Conclusions: it would be recommended that conclusions should be modified according to the changes of the text after answering on general comments. They have to be consistent: a) it was not clear how the portal protein “influences DNA spooling inside the head”; b) A structure of the stopper proteins within fully packaged capsid was not shown: DNA goes through the entire neck channel up to the upper part of the tail, so it does not block the neck channel exit with their gating loops, the structure with the blocked channel was not provided.

A: The effect of anchoring the genome end to the portal on coaxial DNA spooling is discussed on lines 277-279 and illustrated in Fig. 8A-C. We modified the description of the role of stopper proteins in tailless particles to make clearer the fact that it is a hypothesis: “The stopper proteins in the closed conformation block the central channel with their gating loops, suggesting a mechanism for securing the genome inside tailless heads” (lines 392-394).

Minor comments:

Lines 60-62? What was the strategy for selecting phages from cryo-micrographs for processing of neck intermediate structures? Did the authors also study empty phage neck structures (where the genome is fully released)?

A: Please see Fig. S14 for the particle selection and reconstruction strategy. The dataset with genome ejection intermediates included a continuous range of particles with decreasing genome content. We show that residual DNA is present even in particles that 2D classification of heads indicates as empty (to illustrate this, we added a new scheme to Fig. S14 as upper right inset).

We could not determine the structure of the empty particle because we could not be sure that we have a genuine class of empty particles. To clarify how genome release

intermediates were produced and selected, we included the following text into the manuscript (lines 179-182): “We induced genome release in purified phage 812 virions by in vitro treatment with urea. The capsids showed a continuum of residual genome density, as the mechanisms that ensure completion of DNA ejection in vivo cannot occur. We selected capsids with the lowest average inner density, and we termed this dataset the genome release intermediates.”

Lines 71-72: Clarification is needed on which ‘state’ of phage is this structure derived from. Presumably the genome filled state?

A: Thank you, it was the genome-filled virion. We now included this clarification in the section heading: “Neck structure of phage 812 virion” (line 72).

Lines 81-84: Did the authors resolve the entire portal complex as usually the crown domain is less well resolved? One note would be to show a monomer structural model with different parts labelled (with amino acids of N and C termini) to show resolvability and structural conservation (with related phage portal proteins) of the different domains?

A: For the virion portal protein, we built residues 49-378, 395-503 out of 563. To show resolvability, we have now included the local resolution map of the virion neck as new Fig. S5.

Portal protein of phage 812 is the only portal with known structure within its family. We have made a comparison to AlphaFold-predicted portal structures of related phages in Fig. S8.

Line 177: Figure 1D is not obvious that is from the genome release intermediate neck apparatus.

A: We have now included labels of virion and genome release intermediate in Fig. 1.

Also fig. S14A-B do not show the described disordered regions of stopper protein = a cut-away view in S14B would be useful. A figure to show this shift would be helpful.

A: The disordered regions of stopper proteins were not resolved in the cryo-EM map of the genome release intermediate, and we did not build their structures. Therefore, we cannot show them. The comparison of the structures of stopper proteins in 812 virion and genome release intermediate are shown in Fig 1D and described on lines 184-188. The structures are near identical, except for the disordered termini of the stopper protein from the genome release intermediate.

Lines 180-181: The RMSD value seems smaller than the stated resolution. Are these changes significant?

A: The stated RMSD value does not describe the difference in the positioning of the crown domains relative to the portal core in virion vs genome release intermediate. The low RMSD value after crown domain alignment indicates that the structures of portal crowns in virion and genome release intermediate are nearly identical; however, they have different positions relative to the portal core (Fig. 1D). The RMSD of portal proteins in virion vs genome release intermediate, including the crown, is 6.1 Å (we have now included this value in the manuscript as follows (lines 188-191): “The crown domains of portal proteins are shifted 8.9 Å away from the core of the portal complex and rotated 28° counterclockwise around the portal central axis, when looking from the head center, relative to their position in the virion (Fig. 1D, RMSD of 6.1 Å over 5,256 portal protein C α atom pairs.”).

Lines 186-187: The authors describe a comparison of empty vs. filled phage neck protein complexes showing no difference; This has not been described earlier or shown in the figure.

A: Thank you, we incorrectly stated that we compared the virion to the empty particle. Instead we compare and show the superposition of the structures of virion and genome release intermediate in Fig. 1D. This has now been corrected (lines 197-200): “The absence of structural differences between the neck proteins of the phage 812 genome-containing

particles and genome release intermediates indicates that no conformational change to the neck is involved in the regulation of phage 812 genome release”.

What about the comparison of empty vs. genome-release intermediate? Given also the authors describe a DNA kink in the genome release intermediate at adaptor protein and a second DNA genome observed on the wing domain of portal complex in both genome-filled and genome release intermediate, some structural changes would be expected when the DNA is no longer there in both cases?

A: We do not have the structure of the empty 812 particle. We show that residual DNA is present even in particles that 2D classification of heads indicates as empty (Fig. S14 upper right inset), therefore we could not ensure that individual particles do not contain residual DNA.

Line 204: Fig.S17 – the authors show the asymmetric map (in teal) and 12-fold symmetrised map in grey. How is the double-stranded DNA density well resolved (with major and minor grooves clearly well resolved?) in the grey map if the 3D map calculated was symmetrised by applying C12 symmetry? Are the authors implying the DNA is also 12-fold symmetrically arranged? Kindly elaborate.

A: Indeed, the DNA encircling the portal complex in the genome release intermediate binds in such a way that its backbone arrangement matches the twelvefold symmetry of the portal complex. This is explained in section: “Binding of the DNA to the outer surface of the portal complex”, lines 234-237.

Lines 205-207: Isn’t this encircled dsDNA segment also present in the genome-filled (virion) neck complex – albeit not well resolved in the symmetrised structure (Figure 1), if so, what are the differences between ‘states’ of this dsDNA.

A: There are three rings of DNA encircling the portal complex of 812 virion (Figs. 5A and S5A). However, the diameters of the DNA circles in 812 virion are such that the helical periodicity of the DNA does not match that of the twelvefold-symmetric portal complex. We have now included a new Table S5 with parameters of the DNA rings.

“Table S5. Interactions between portal complex and DNA rings in virion and genome release intermediate. Distance d reports the minimal distance between a phosphate oxygen of a B-form circular dsDNA model and a charged nitrogen of an Arg or Lys side chain of the portal dodecamer.”

Particle type	dsDNA ring radius (Å)	modelled DNA length (bp)	$d \leq 4 \text{ \AA}$	$4 \text{ \AA} < d \leq 6 \text{ \AA}$
virion	52	94	Arg137, Lys465	Lys138
	71	131	Lys138, Lys194	Arg145, Lys425
	85	155	Lys189, Lys192, Lys212, Lys221	
genome release intermediate	65	120	Arg137, Arg145, Lys425	Lys194

Lines 209-211. Please explain the following sentence (it sounds somewhat confusing): “...the DNA double helix makes one turn per portal monomer...”. What is the conformation of this circular dsDNA around the portal complex?

A: The arc length between the neighboring portal proteins at the distance of 65 Å from the portal axis at which the DNA encircles the portal of the genome release intermediate is 34

Å, which corresponds to one turn of B-form DNA helix. We have now depicted this in Fig. 4B, with caption: “The arc distance between neighboring wing domains described by the anchor DNA is 34 Å (indicated in navy blue), which corresponds to one B-DNA helix turn.”.

Given the state that has been captured of the portal complex with encircled DNA at wing and crown domain. Can the authors comment on any local conformational changes at the portal which may be present (when compared with portal proteins of similar phylogeny) to accommodate the encircled dsDNA?

A: There are no experimentally determined structures of portal proteins of related phages. However, a comparison of AlphaFold-predicted structures of 812-related phages is presented in Fig. S8. The positive charges that enable 812 portal complex to bind DNA are preserved in related phages but absent in more distant phages (Fig. S9).

Line 218-219: What do the authors mean by ‘inter-particle’ averaging?

A: The single-particle reconstruction process uses projection images of different particles that have identical structures to determine the three-dimensional structure of the macromolecular complex (Figs. S1 and S14). In case the particles differ in small details, such as nucleotide identity, there is not sufficient signal to distinguish them and the resulting three-dimensional reconstruction represents the average of the structures present in the dataset of two-dimensional projection images.

Did the authors try masked classification to see if they can separate different subsets of particles to improve DNA density?

A: We used masked classification in the reconstruction of the DNA encircling the portal complex of genome release intermediate to determine the position of the circle opening (Fig. 4A-B). However, the quality of data and current reconstruction algorithms is not sufficient to classify megadalton macromolecular complexes based on differences in individual nucleotides.

Based on the DNA mapping of the segment bound to portal – can the authors comment on whether this is sub-fragment of DNA that is independent to the genome packaged inside the capsid?

A: There is no evidence that 812 particles contain segments of DNA independent of the genome. Our analysis of DNA residual in 812 genome release intermediates demonstrates that the DNA remaining in the particles corresponds to the left long terminal repeat (Fig. 4G).

How are the two connect as based on the structures they seem independent (both in virion and genome-release intermediate?).

A: Our cryo-EM reconstructions resolve only short stretches of DNA that closely interact with portal complex. However, these DNA stretches are part of phage 812 genomic DNA as evidenced by our sequencing analysis of residual DNA in genome release intermediates.

Lines 254-255. English has to be corrected: "...loop interacts with the crown base of the same portal protein (explain what is meant by "the same portal protein?"). Are the authors discussing interactions within one subunit?

A: Yes, we describe interactions within one subunit. We have now modified the sentence to clarify this point (line 282): "Each tunnel loop interacts with the crown base of the same portal protein subunit".

Line 284-296 – The bent DNA at portal-adaptor junction also present in the virion map (Fig. S19 B); do the authors imply that in two states of the phage, the DNA is stalled in a similar way whereby the identity of the DNA segment present in virion and genome-release is the same?

A: The structures of the DNA inside the adaptor channel are similar (Fig. S23); however, the DNA segments in the virion and the genome release intermediate are different. In the virion the DNA corresponds to the end of the right long terminal repeat (discussed on lines 353-360 and shown in Fig. 7C). In contrast, in genome release intermediates the adaptor channel contains various sequences based on the stage of genome ejection in a particular particle.

What would be the biological reason for having a stalled DNA state in the virion (as DNA is fully packaged in the capsid)?

A: The formation of A-form DNA inside the adaptor channel may help to retain the genome inside the virion as the pressure of the packaged genome is distributed between the tail tape measure proteins and, through the off-centric DNA, the adaptor complex (discussed on lines 324-329).

Line 317. What is it "the right end of phage 812"., The term "right" is unclear , it is confusing this given context.

A: To describe the genome of phage 812, we use the nomenclature established for Kayviruses by Łobocka, 2012. The global orientation of the genome is chosen based on the predominant orientation of its coding sequences. The packaged genome contains direct "long terminal repeats" at each end (Botka, 2019). We included the following clarification in the manuscript (lines 254-256): "The repeats are conventionally named as left, the repeat at the 5' end of the genomic sequence, and right, the 3' end repeat downstream of the genomic sequence (Łobocka, 2012)".

** See the Nature Portfolio author and referees' website at www.nature.com/authors for information about policies, services and author benefits

Communications Biology is committed to improving transparency in authorship. As part of our efforts in this direction, we are now requesting that all authors identified as 'corresponding author' create and link their Open Researcher and Contributor Identifier (ORCID) with their account on the Manuscript Tracking System prior to acceptance. ORCID helps the scientific community achieve unambiguous attribution of all scholarly contributions. You can create and link your ORCID from the home page of the Manuscript Tracking System by clicking on 'Modify my Springer Nature account' and following the instructions in the link below. Please also inform all co-authors that they can add their ORCIDs to their accounts and that they must do so prior to acceptance.

If you experience problems in linking your ORCID, please contact the Platform Support Helpdesk.

This email has been sent through the Springer Nature Tracking System NY-610A-NPG&MTS

Confidentiality Statement:

This e-mail is confidential and subject to copyright. Any unauthorised use or disclosure of its contents is prohibited. If you have received this email in error please notify our Manuscript Tracking System Helpdesk team at <http://platformsupport.nature.com> .

Details of the confidentiality and pre-publicity policy may be found here <http://www.nature.com/authors/policies/confidentiality.html>

Privacy Policy | Update Profile

Note: line numbers refer to the tracked version of the manuscript (all markup)

Response to Reviewer #2

Minor points:

Line 187-190: 'The crown of the portal complex moves as a rigid body (RMSD of 1.4 Å over 804 Cα atom pairs after crown alignment).' Based on the cryo-EM derived model (from 4 Angstrom map), reporting a 'change' of 1.4 Å does not seem comparable on the level of cryoEM derived model. The authors can simply say 'minimal change was observed when crown domain models are compared.'

A: The suggestion was accepted; the cited sentence was replaced by the following text: "Aside from the change in position, enabled by flexible linkers connecting crown and wing domains of portal proteins, minimal difference was observed when crown domain models of virion and genome release intermediate were compared." (lines 187-192).

The authors also later mention the crown funnel tilts by 3.5 degrees (Line 261-262), based on interactions with DNA loop. I think some clarification would help make this point strong that some local changes are induced but no large changes were observed as a result of DNA present in the channel.

A: To clarify that crown tilt does not involve a change in the crown funnel structure but is instead enabled by flexible linkers to portal wings, we modified the text to read: "The crown funnel, connected to the core of the portal complex by flexible loops, is tilted 3.5° off the neck axis [...]" (lines 262-263).

Given no major changes were observed between genome containing and genome-release intermediate, it can be attributed to the 'stalled' state of the complex?

A: The observation that structures of neck proteins stay unchanged by genome ejection – apart from differences localized to portal tunnel loops and/or crown – was noted in multiple phages (Bárdy, 2020 <https://doi.org/10.1038/s41467-020-16669-9>; Chen 2021, <https://doi.org/10.1073/pnas.2102003118>; Valentová, 2024 <https://doi.org/10.1038/s44318-024-00195-1>). Therefore, we do not attribute the lack of major differences between phage 812 neck structures of genome containing virion and the genome release intermediate to the presence of stalled DNA. In phages with contractile tails such as 812, tail sheath contraction was hypothesized to induce changes in the neck structure, but this appears to not be the case (this work, lines 219-225, and Li, 2023 <https://doi.org/10.1038/s41467-023-39756-z>, discussed on lines 225-228).